# Why Low-Precision Transformer Training Fails: An Analysis on Flash Attention

**Haiquan Qiu**[1]     **Quanming Yao**[1,2,3] [*]
[1]Department of Electronic Engineering, Tsinghua University
[2]Beijing National Research Center for Information Science and Technology
[3]State Key laboratory of Space Network and Communications
`qhq22@mails.tsinghua.edu.cn, qyaoaa@tsinghua.edu.cn`

## Abstract

The pursuit of computational efficiency has driven the adoption of low-precision formats for training transformer models. However, this progress is often hindered by notorious training instabilities. This paper provides the first mechanistic explanation for a long-standing and unresolved failure case where training with flash attention in low-precision settings leads to catastrophic loss explosion. Our in-depth analysis reveals that the failure is not a random artifact but caused by two intertwined phenomena: the emergence of similar low-rank representations within the attention mechanism and the compounding effect of biased rounding errors inherent in low-precision arithmetic. We demonstrate how these factors create a vicious cycle of error accumulation that corrupts weight updates, ultimately derailing the training dynamics. To validate our findings, we introduce a minimal modification to the flash attention that mitigates the bias in rounding errors. This simple change stabilizes the training process, confirming our analysis and offering a practical solution to this persistent problem. Code is available at `https://github.com/ucker/why-low-precision-training-fails`.

## 1 Introduction

The pursuit of training ever-larger and more powerful transformer models is a relentless drive for computational efficiency (Brown et al., 2020; Hoffmann et al., 2022). A key strategy in this endeavor is the adoption of low-precision numerical formats (Micikevicius et al., 2017; Wang et al., 2018; Kalamkar et al., 2019; Liu et al., 2024), which promise substantial reductions in memory footprint and significant boosts in training speed. In industrial practice, it is common to use BF16 for memory-bound operations like flash attention while pushing compute-bound operations like FFNs to even lower precisions such as FP8 (Liu et al., 2024; Qwen-Team, 2025). This highlights the heightened sensitivity of attention mechanisms to numerical precision. Despite the development of stabilization techniques like QK normalization (Henry et al., 2020; Qwen-Team, 2025), QK-clip (Kimi-Team, 2025) and Gated Attention (Qiu et al., 2025b; Qwen-Team, 2025), the path to further reducing precision is often blocked by a lack of understanding of the underlying failure mechanisms.

This paper confronts this challenge by dissecting a notorious and long-standing failure issue involving flash attention. By reducing the memory complexity of the attention mechanism from quadratic to linear with respect to sequence length, flash attention has become a cornerstone algorithm for efficient transformer training, making it indispensable for handling the long contexts required by modern large-scale models (Dao et al., 2022; Dao, 2024; Shah et al., 2024). This failure, arising in low-precision settings (flash-attention Issue 337, 2024; nanoGPT Issue 303, 2023; nanoGPT Issue 524, 2024; nanoGPT Issue 554, 2024; Lee et al., 2024; Golden et al., 2024), presents a significant bottleneck. We focus on a specific, reproducible failure case reported by the community (nanoGPT Issue 303, 2023; nanoGPT Issue 524, 2024), which has remained unresolved for over two years. Our in-depth analysis provides the first mechanistic explanation for this failure, revealing that it is not a random artifact but a direct consequence of two intertwined phenomena: the emergence of similar low-rank representations across different training steps and tokens, and the compounding effect

---

[*]Corresponding author

of biased rounding errors inherent in low-precision arithmetic. We demonstrate how these biased rounding errors act as coefficients for the low-rank representations, causing them to accumulate as a biased gradient update to the weights. This pushes the spectral norm of weights and activations to increase abnormally, ultimately overwhelming the training dynamics. To validate our analysis, we introduce a minimal modification to flash attention that mitigates the bias in rounding errors, allowing the low-rank weight updates to cancel out during training. This experiment confirms our analysis and stabilizes the training process, offering a practical solution to this persistent problem.

**Notations** We use bold lowercase letters for vectors (e.g., $\boldsymbol{\delta}, \mathbf{m}, \mathbf{r}_m$) and bold uppercase letters for matrices (e.g., $\mathbf{Q}, \mathbf{K}, \mathbf{V}$). We use notation $d\mathbf{W}$ to denote the gradient of the loss $\ell$ with respect to variable $\mathbf{W}$ for simplicity, e.g., $d\mathbf{Q} := d\ell/d\mathbf{Q}$. $\mathrm{diag}(\mathbf{v})$ denotes a diagonal matrix with the elements of vector $\mathbf{v}$ on its diagonal. $\circ$ denotes the element-wise product. We use Python-style indexing, e.g., $\mathbf{M}[i,:]$ denotes the $i$-th row of matrix $\mathbf{M}$. Subscripts $lp$ and $hp$ distinguish between low-precision (BF16) and high-precision (FP32) computations. Binary strings are represented using a typewriter font (e.g., 101010). The where function mimics torch.where. Unless specified otherwise, operations follow PyTorch's broadcasting rules. Key claims are highlighted in light cyan box at the end of some sections.

## 2 PRELIMINARY

### 2.1 LOW-PRECISION TRAINING

Low-precision training is a cornerstone of modern deep learning, enabling the development of increasingly large models by reducing memory usage and accelerating computation (Liu et al., 2024; Tseng et al., 2025). This is achieved by representing weights, activations, and gradients using numerical formats with fewer bits than the standard 32-bit single-precision (FP32). Mixed-precision training (Micikevicius et al., 2017) has been widely adopted in practice, which combines 16-bit formats like FP16 or bfloat16 (BF16) for most computations with an FP32 master copy of weights to maintain accuracy. While FP16 offers higher precision, its limited dynamic range often leads to gradient underflow, requiring techniques like loss scaling. In contrast, BF16, originally developed for Google's TPUs and now widely supported, provides the same dynamic range as FP32, making it more robust against underflow and a preferred choice for training large language models (Kalamkar et al., 2019; Wang & Kanwar, 2019). However, the reduced precision of BF16 can still introduce numerical errors that lead to training failure, the focus of this paper.

The bfloat16 (BF16) format is a 16-bit floating-point representation with 1 sign, 8 exponent, and 7 significand bits. It matches the dynamic range of 32-bit single-precision (FP32) but has lower precision, making it a popular choice for balancing computation and numerical range in deep learning. Adding two BF16 numbers involves aligning their exponents, adding the significands, normalizing the sum, and rounding the result to fit the 7-bit fraction. This final rounding step, typically "round to nearest, ties to even", is one of the primary sources of error. While this rounding method is designed to be unbiased for random data, a sequence of operations on data with a specific distribution could lead to a *biased rounding error*. This accumulation of error in one direction is a critical factor to the training failure observed in low-precision settings. See Appendix B for details of BF16 addition.

### 2.2 FLASH ATTENTION

Flash Attention (FA) (Dao et al., 2022; Dao, 2024; Shah et al., 2024) is an I/O-aware exact attention algorithm designed to overcome the memory bottleneck of standard attention. Standard attention, defined as $\mathbf{O} = \mathrm{softmax}(\alpha \mathbf{Q}\mathbf{K}^\top)\mathbf{V}$, requires materializing the $N \times N$ attention score matrix $\mathbf{S} = \alpha \mathbf{Q}\mathbf{K}^\top$, leading to a memory complexity of $\mathcal{O}(N^2)$ with respect to the sequence length $N$. Flash attention reduces this to $\mathcal{O}(N)$ by partitioning the input matrices $\mathbf{Q}, \mathbf{K}, \mathbf{V} \in \mathbb{R}^{N \times d}$ into blocks and processing them iteratively. These blocks are loaded from high-bandwidth memory (HBM) into fast on-chip SRAM, minimizing costly memory transfers.

In this paper, we focus on analyzing flash attention 2. The forward pass of FA computes the output $\mathbf{O}$ and log-sum-exp statistics $\mathbf{L}$ using an online softmax method (Algorithm 1). It iterates through blocks of $\mathbf{Q}$ (outer loop) and blocks of $\mathbf{K}, \mathbf{V}$ (inner loop). For each query block $\mathbf{Q}_i$, it maintains running statistics: the maximum score $\mathbf{m}_i$ and the normalization factor $\boldsymbol{\ell}_i$. In each inner loop step,

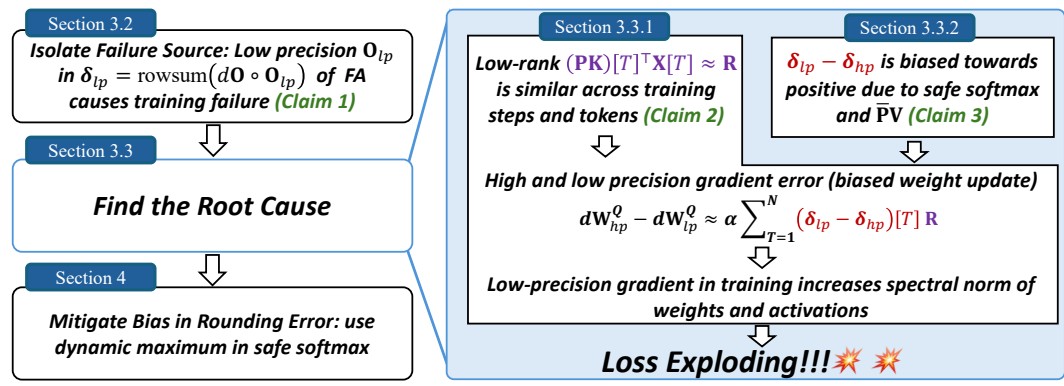

Figure 1: Analysis in different sections. Our paper traces the causal chain of training failure (blue box) in reverse to identify the root causes.

it computes unnormalized attention scores $\bar{\mathbf{P}}_i^{(j)} = \exp(\mathbf{S}_i^{(j)} - \mathbf{m}_i^{(j)})$ and updates an unnormalized output by accumulating the product $\bar{\mathbf{P}}_i^{(j)}\mathbf{V}_j$. This accumulation is a key focus of our analysis. After iterating through all key/value blocks, the final output block $\mathbf{O}_i$ is correctly normalized. This tiling strategy avoids materializing the full $N \times N$ score matrix. The backward pass (Algorithm 2) leverages the same tiling strategy. *It first computes a key intermediate term, $\delta = \mathrm{rowsum}(d\mathbf{O} \circ \mathbf{O})$, which is central to our investigation.* Then, it recomputes attention scores on-the-fly to calculate the gradient of the scores, $d\mathbf{S}_i^{(j)} = \mathbf{P}_i^{(j)} \circ (d\mathbf{P}_i^{(j)} - \delta_i)$, where $d\mathbf{P}_i^{(j)} = d\mathbf{O}_i\mathbf{V}_j^\top$. The final gradients $d\mathbf{Q}, d\mathbf{K}, d\mathbf{V}$ are accumulated block-wise. This approach maintains I/O efficiency in both passes.

## 3 ROOT CAUSES OF INSTABILITY IN FLASH ATTENTION

We first introduce the failure case in Section 3.1. In Section 3.2, we narrow down the source of the numerical errors within the flash attention. Further analysis in Section 3.3 reveals that the failure stems from a combination of two factors: the emergence of low-rank representations and the accumulation of biased rounding errors inherent to BF16 arithmetic. The full process of such failure from root cause to loss exploding is shown in Fig. 1.

### 3.1 THE FAILURE CASE OF LOW-PRECISION FLASH ATTENTION

Our investigation targets a well-documented and persistent failure: the catastrophic loss explosion that occurs when training Generative Pre-trained Transformer 2 (GPT-2) models with flash attention in BF16 precision (nanoGPT Issue 303, 2023; nanoGPT Issue 524, 2024; nanoGPT Issue 554, 2024). This failure case, reported for over two years, manifests as a sudden loss explosion after several thousand training steps (see Fig. 10). While empirical workarounds like reverting to standard attention or using higher precision (FP32) stabilize training, they come at the cost of efficiency. This instability is not an isolated incident; the broader community has observed similar failures when training large language models (Kimi-Team, 2025; Qwen-Team, 2025). These failures are often empirically linked to phenomena such as large spectral norms of weights, large activations (Yang

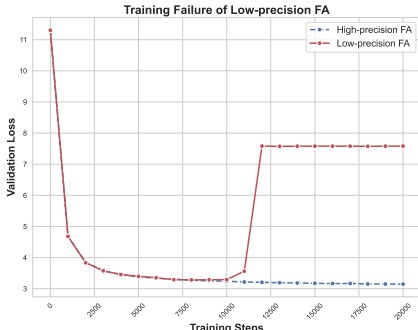

Figure 2: The failure case using BF16 and flash attention results in a sudden loss explosion, while the stable configuration converges.

et al., 2023; Rybakov et al., 2024), and attention sinks (Xiao et al., 2023), leading to a suite of fixes including QK normalization (Henry et al., 2020; Qwen-Team, 2025), QK-clipping (Kimi-Team, 2025), and Gated Attention (Qiu et al., 2025b; Qwen-Team, 2025). Despite these interventions, a fundamental understanding of the root causes has remained elusive. The absence of a clear causal chain from numerical error to loss explosion has left the community reliant on ad-hoc patches rather

than principled solutions, hindering progress in robust low-precision training. This paper provides the first mechanistic explanation by reproducing the failure, dissecting its root causes, and proposing a practical, principled solution.

To reproduce the failure, we employ a GPT-2 architecture with 12 layers, 12 attention heads, an embedding dimension of 768, and a context length of 1024. The model is pre-trained on the OpenWeb-Text dataset (Gokaslan et al., 2019). For deterministic reproducibility, we deviate from a standard random data loader by recording and reusing the exact sequence of data batches from an initial run that led to the failure. This ensures all subsequent experiments process identical data in the same order, isolating the failure from data-related randomness.

We train the model using the AdamW optimizer with $\beta_1 = 0.9$, $\beta_2 = 0.95$, and zero weight decay. The learning rate follows a cosine schedule with a 2000-iteration linear warmup to a peak of $1 \times 10^{-3}$, decaying to $1 \times 10^{-5}$. We apply global gradient clipping with a maximum norm of 1.0. Training is conducted on 4 NVIDIA A100 (80GB) GPUs using PyTorch's Distributed Data Parallel (DDP) module. We use automatic mixed precision with BF16 for the forward pass and FP32 for the backward pass. Each GPU processes a micro-batch size of 32, and with gradient accumulation over 4 steps, the effective global batch size is 524,288 tokens per optimization step.

## 3.2 ISOLATING THE SOURCE OF FAILURE WITHIN FLASH ATTENTION

To pinpoint the source of failure within flash attention, we conduct a series of targeted experiments. We systematically modify the algorithm—disabling tiling, selectively replacing flash attention with a standard implementation, and performing key computations in high precision—to narrow down the potential causes of instability. To accelerate this analysis, we monitor spectral alignment (Qiu et al., 2025a) to quickly identify failing configurations.

**Tiling is not the Source of Failure.** To determine if the block-wise processing in flash attention was responsible for the failure, we conduct an experiment where we disabled tiling by setting the block size equal to the sequence length. This forces the algorithm to compute with the full matrices at once. The training process still failed, leading to the same loss explosion. This finding rules out the tiling strategy as the cause of the problem. For all subsequent experiments, we therefore use this non-tiled setup to simplify the analysis and focus on the core numerical computations.

**Failure Originates in a Single Layer.** We first analyze the spectral norms of weights across all layers (Yang et al., 2023; Rybakov et al., 2024). This reveals an anomalous spike specifically within the second layer's attention (see Fig. 11). We confirm this finding with two targeted experiments: (1) using flash attention only in layer 2 was sufficient to reproduce the training failure, and (2) replacing flash attention with standard attention in layer 2, while retaining it in all other layers, restored training stability. These results conclusively identify the flash attention in layer 2 as the origin of the failure. So, subsequent analysis focuses on this module to dissect the failure mechanism.

**Failure is Linked to the Computation of $\delta$.** The backward pass of flash attention uses the term $\delta = \mathrm{rowsum}(d\mathbf{O} \circ \mathbf{O}) \in \mathbb{R}^N$ for computational efficiency. An alternative, mathematically equivalent formulation computes this term as $\delta = \mathrm{rowsum}(d\mathbf{P} \circ \mathbf{P})$, where $d\mathbf{P} = d\mathbf{O}\mathbf{V}^\top$. We find that replacing the efficient computation with this alternative formulation restored training stability. This experiment demonstrates that numerical errors introduced when computing $\mathbf{O}$ in BF16 is likely the primary source of failure, as the alternative formulation which avoids training failure is equivalent to use $\mathbf{O}$ computed in FP32.

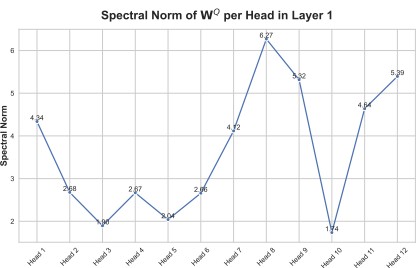

Figure 3: $\mathbf{W}^Q$ of attention head 8 has the largest spectral norm. Subsequent analysis focuses on this head.

**Numerical Errors in $\mathbf{O}$ are the Source of Failure.**
Building on the finding that the computation of $\delta$ is critical, we further isolate the source of error to the output matrix $\mathbf{O}_{lp}$ in low-precision $\delta_{lp} = \mathrm{rowsum}(d\mathbf{O} \circ \mathbf{O}_{lp})$. We conduct two key experiments. First, instead of using the low-precision $\mathbf{O}$ from the forward pass to compute $\delta$, we recompute it as $\mathbf{PV}$ in FP32 within the backward pass; this change stabilizes training. Second, we find that computing $\mathbf{O}$ in high precision (FP32) during the forward pass, i.e., $\delta_{hp} = \mathrm{rowsum}(d\mathbf{O} \circ \mathbf{O}_{hp})$, while

keeping all other operations in BF16, also restored stability. This evidence conclusively demonstrates that numerical errors introduced during the BF16 computation of $\mathbf{O}$ are the direct cause of the failure (refer to Claim 1).

**Failure is Localized to Specific Attention Heads.** To further narrow down the source of failure, we analyze individual attention heads by tracking the spectral norm (Yang et al., 2023; Rybakov et al., 2024) of their query projection matrices ($\mathbf{W}^Q$), as shown in Fig. 3. This reveals that a few heads exhibit disproportionately large spectral norms. We confirm their role by selectively computing the output $\mathbf{O}$ in high precision for these outlier heads (1, 7, 8, 9, 11, and 12), which is sufficient to restore training stability. *Since head 8 shows the largest spectral norm, we focus our subsequent analysis on this head to dissect the precise failure mechanism.*

> **Claim 1.** *Low-precision $\boldsymbol{\delta}_{lp} = \mathrm{rowsum}(d\mathbf{O} \circ \mathbf{O}_{lp})$ causes training failure.*

### 3.3 THE ROOT CAUSES OF TRAINING FAILURE

Our investigation in this section uncovers the two interconnected root causes of the training failure. In Section 3.3.1, we demonstrate how low-precision $\boldsymbol{\delta}_{lp}$ drives the training failure by a biased weight update, and find that bias arises from the emergence of similar low-rank representations $\mathbf{R}$, whose coefficients $(\boldsymbol{\delta}_{lp} - \boldsymbol{\delta}_{hp})[T]$ are biased towards positive values, causing the error to accumulate rather than cancel out. In Section 3.3.2, we trace the origin of these positive coefficients $(\boldsymbol{\delta}_{lp} - \boldsymbol{\delta}_{hp})[T]$ to biased rounding errors inherent in the BF16 addition within the $\bar{\mathbf{P}}\mathbf{V}$ product.

#### 3.3.1 CAUSE 1. SIMILAR LOW-RANK MATRICES BIAS WEIGHT UPDATES

This section traces the training failure to biased weight update. We first analyze the difference between the high and low precision gradients calculated with $\boldsymbol{\delta}_{hp}$ and $\boldsymbol{\delta}_{lp}$, respectively. Then we find that the low-rank representations and biased $(\boldsymbol{\delta}_{lp} - \boldsymbol{\delta}_{hp})[T]$ lead to loss explosion.

**Analysis of the Error between High and Low Precision Gradients** To understand how numerical errors propagate into the gradients, we analyze the difference between the high-precision ($hp$) and low-precision ($lp$) gradients for the query matrix, $d\mathbf{Q}$. The gradient $d\mathbf{Q}$ is computed from the gradient of the attention scores, $d\mathbf{S}$, as $d\mathbf{Q} = d\mathbf{S}\mathbf{K}$. The score gradient is given by $d\mathbf{S} = \alpha\mathbf{P} \circ (d\mathbf{P} - \boldsymbol{\delta})$, where $\boldsymbol{\delta} = \mathrm{rowsum}(d\mathbf{O} \circ \mathbf{O})$ is the only term that differs between the high- and low-precision backward passes based on Section 3.2, and $\alpha$ is a scaling factor in attention.

The difference between the high-precision and low-precision query gradients can be derived as:

$$\begin{aligned}
d\mathbf{Q}_{hp} - d\mathbf{Q}_{lp} &= (d\mathbf{S}_{hp} - d\mathbf{S}_{lp})\mathbf{K} \\
&= \left(\alpha\mathbf{P} \circ (d\mathbf{P} - \boldsymbol{\delta}_{hp}) - \alpha\mathbf{P} \circ (d\mathbf{P} - \boldsymbol{\delta}_{lp})\right)\mathbf{K} = \left(\alpha\mathbf{P} \circ (\boldsymbol{\delta}_{lp} - \boldsymbol{\delta}_{hp})\right)\mathbf{K} \\
&= \alpha \cdot \mathrm{diag}(\boldsymbol{\delta}_{lp} - \boldsymbol{\delta}_{hp})(\mathbf{P}\mathbf{K}).
\end{aligned} \tag{1}$$

In the final step, we express the row-wise scaling operation $\mathbf{P} \circ (\boldsymbol{\delta}_{lp} - \boldsymbol{\delta}_{hp})$ (follow broadcasting rule) as a matrix multiplication, where $\mathrm{diag}(\boldsymbol{\delta}_{lp} - \boldsymbol{\delta}_{hp})$ is a diagonal matrix whose diagonal entries are the elements of the vector difference $\boldsymbol{\delta}_{lp} - \boldsymbol{\delta}_{hp}$. This formulation reveals that the gradient error is directly proportional to the error in $\boldsymbol{\delta}$ and is modulated by the term $\mathbf{P}\mathbf{K}$.

The gradient of the query projection matrix, $d\mathbf{W}^Q$, is given by the outer product of the input features $\mathbf{X}$ and the query gradient $d\mathbf{Q}$. The difference between the high-precision ($hp$) and low-precision ($lp$) gradients for $\mathbf{W}^Q$ can be expressed as:

$$\begin{aligned}
d\mathbf{W}^Q_{hp} - d\mathbf{W}^Q_{lp} &= (d\mathbf{Q}_{hp} - d\mathbf{Q}_{lp})^\top\mathbf{X} = \alpha(\mathbf{P}\mathbf{K})^\top\mathrm{diag}(\boldsymbol{\delta}_{lp} - \boldsymbol{\delta}_{hp})\mathbf{X}, \\
&= \alpha\sum_{T=1}^{N}(\boldsymbol{\delta}_{lp} - \boldsymbol{\delta}_{hp})[T] \cdot (\mathbf{P}\mathbf{K})[T]^\top\mathbf{X}[T],
\end{aligned} \tag{2}$$

where $(\mathbf{P}\mathbf{K})[T]$ and $\mathbf{X}[T]$ are the $T$-th row vectors of their matrices. This equation shows that the total gradient error is a weighted sum of rank-1 matrices, with weights given by the error in $\boldsymbol{\delta}$.

**Similar Low-rank Updates of Weight Cause Training Failure** In Fig. 4, the rows of $\mathbf{P}\mathbf{K}$ (panels a, d) and $\mathbf{X}$ (panels b, e) exhibit strong structural similarity across different training steps and token

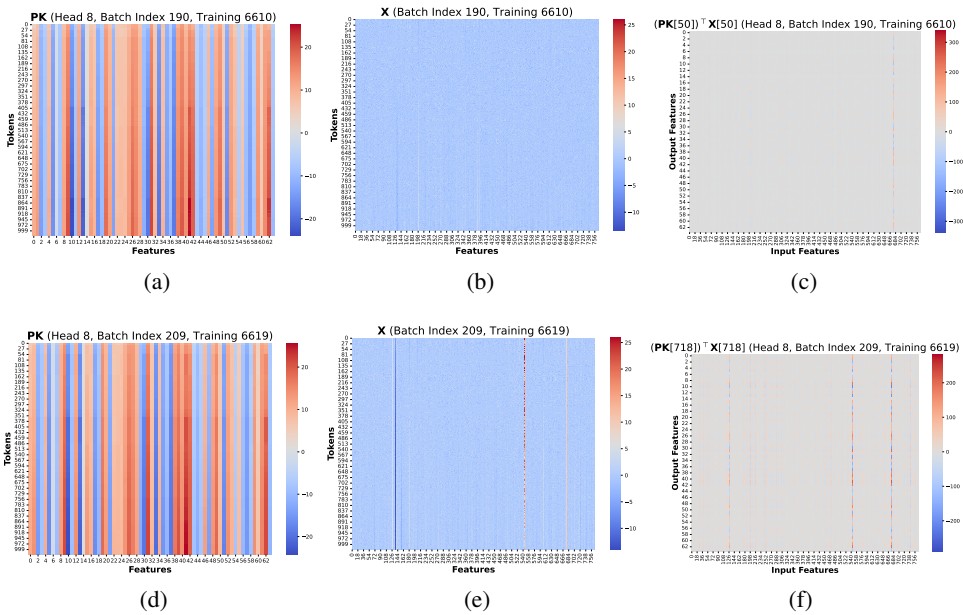

Figure 4: $\mathbf{PK}$, $\mathbf{X}$, and $(\mathbf{PK})[T]^\top\mathbf{X}[T]$ at different batch indices and training steps. (c) and (f) show that $(\mathbf{PK})[T]^\top\mathbf{X}[T]$ for different tokens and training steps have some similar columns in input features 546 and 678.

positions. This implies that the resulting rank-1 matrices, $(\mathbf{PK})[T]^\top\mathbf{X}[T]$, are also highly similar to one another. For instance, Fig. 4 (panels c, f) shows this similarity for tokens 50 and 718 at training steps 6610 and 6619, respectively. Because these rank-1 error components are structurally consistent, we can approximate the total gradient difference as

$$d\mathbf{W}_{hp}^Q - d\mathbf{W}_{lp}^Q \approx \alpha \sum_{T=1}^{N} (\boldsymbol{\delta}_{lp} - \boldsymbol{\delta}_{hp})[T]\mathbf{R} \tag{3}$$

where $\mathbf{R}$ denotes the common low-rank structure emerging across different tokens and training steps.

Eqn.(3) shows that the accumulation of the low-rank error direction $\mathbf{R}$ is governed by the scalar term $\sum_{T=1}^{N}(\boldsymbol{\delta}_{lp}-\boldsymbol{\delta}_{hp})[T]$. If this sum is biased towards non-zero values, the error across different training steps will accumulate rather than cancel out. We track the cumulative sum of $\sum_{T=1}^{N}(\boldsymbol{\delta}_{lp}-\boldsymbol{\delta}_{hp})[T]$ over a sequence of training steps (6580 to 6680) leading up to the failure, as shown in Fig. 5(a). The plot reveals that this sum is consistently positive, indicating a systematic bias. This bias causes the error in the low-rank direction $\mathbf{R}$ to compound with each training step. Because $\mathbf{R}$ is also similar for different steps, this ultimately corrupts the weight updates, increases the spectral norm (Fig. 11) and activations (Yang et al., 2023; Rybakov et al., 2024), and causes the training failure (refer to Claim 2). The following section finds the root cause of this positive bias by analyzing the weights and gradients at training step 6619, a point of significant positive contribution identified in Fig. 5(a).

> **Claim 2.** *Weight update in low-precision training is biased by $(\boldsymbol{\delta}_{lp} - \boldsymbol{\delta}_{hp})[T]\mathbf{R}$, which arises from the structurally similar matrices (denoted as $\mathbf{R}$) across tokens and training steps, and its positively-biased coefficient $(\boldsymbol{\delta}_{lp} - \boldsymbol{\delta}_{hp})[T]$. This bias accumulates error, preventing cancellation, increasing weight spectral norm and activation, and leading to loss explosion.*

### 3.3.2 CAUSE 2. BIASED ROUNDING ERROR LEADS TO POSITIVE $(\boldsymbol{\delta}_{lp} - \boldsymbol{\delta}_{hp})[T]$

This section investigates the origin of the positive bias in $(\boldsymbol{\delta}_{lp} - \boldsymbol{\delta}_{hp})[T]$. We trace this error to the interaction between $d\mathbf{O}$ and the numerical discrepancy in $\mathbf{O}_{lp} - \mathbf{O}_{hp}$, which itself arises from biased rounding errors in BF16 addition during the $\bar{\mathbf{P}}\mathbf{V}$ computation.

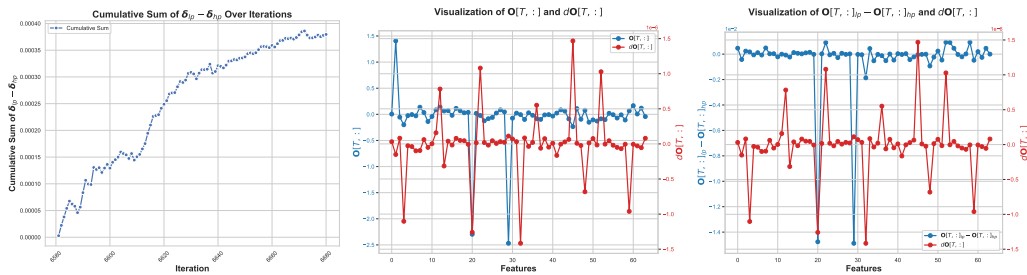

(a) Positively-biased $(\boldsymbol{\delta}_{lp}-\boldsymbol{\delta}_{hp})[T]$ (b) Large values in features 20 & (c) Large error in features 20 & 29
29

Figure 5: Analysis of $\boldsymbol{\delta} = \text{rowsum}(d\mathbf{O} \circ \mathbf{O})$.

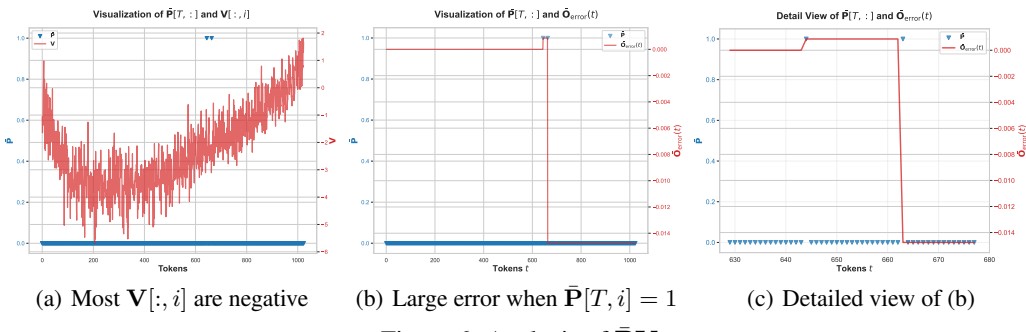

(a) Most $\mathbf{V}[:,i]$ are negative  (b) Large error when $\bar{\mathbf{P}}[T,i]=1$  (c) Detailed view of (b)

Figure 6: Analysis of $\bar{\mathbf{P}}\mathbf{V}$

**Locate the Large Error in** $(\boldsymbol{\delta}_{lp} - \boldsymbol{\delta}_{hp})[T]$  We first investigate the source of the positive bias in $\sum_{T=1}^{N}(\boldsymbol{\delta}_{lp} - \boldsymbol{\delta}_{hp})[T]$. As analyzed in Section 3.2, the error in $\boldsymbol{\delta}$ stems from the product of the upstream gradient $d\mathbf{O}$ and the numerical error in the low-precision output, $\mathbf{O}_{lp} - \mathbf{O}_{hp}$. To dissect this, we focus on a token position $T = 718$ where the error component $(\boldsymbol{\delta}_{lp} - \boldsymbol{\delta}_{hp})[T]$ is positive.

In Fig. 5(b) and (c), we observe a strong sign correlation between the gradient $d\mathbf{O}[T,:]$ and the output error $\mathbf{O}_{lp}[T,:] - \mathbf{O}_{hp}[T,:]$ for specific feature dimensions such as 20 and 29 (also observed across other tokens). In these dimensions, both $d\mathbf{O}$ and the output error $\mathbf{O}_{lp} - \mathbf{O}_{hp}$ are consistently negative. This alignment ensures their product, which contributes to the error in $\boldsymbol{\delta}$, is positive. The fact that the output error tends to be negative ($\mathbf{O}_{lp}[T,i] < \mathbf{O}_{hp}[T,i]$) indicates that the low-precision computation of $\mathbf{O}$ is systematically biased towards more negative values. Our subsequent analysis, therefore, focuses on identifying the origin of this computational bias.

The output $\mathbf{O}$ is computed from an intermediate unnormalized output, $\bar{\mathbf{O}}$. The computation involves a safe softmax followed by a matrix multiplication and normalization:

$$\bar{\mathbf{P}} = \exp(\mathbf{S} - \text{rowmax}(\mathbf{S})), \qquad \bar{\mathbf{O}} = \bar{\mathbf{P}}\mathbf{V}, \qquad \mathbf{O} = \bar{\mathbf{O}}/\text{rowsum}(\bar{\mathbf{P}}).$$

Further experiments pinpoint the source of failure to the computation of the unnormalized output, $\bar{\mathbf{O}} = \bar{\mathbf{P}}\mathbf{V}$. We find that computing only this product in FP32 is sufficient to stabilize training. To understand the origin of this bias, we examine the difference between the low-precision and high-precision computation of a single element, $\bar{\mathbf{O}}[T,i]$ (for feature index $i = 20$ in our analysis):

$$\bar{\mathbf{O}}_{lp}[T,i] - \bar{\mathbf{O}}_{hp}[T,i] = (\bar{\mathbf{P}}_{lp}[T,:]\mathbf{V}[:,i])_{lp} - (\bar{\mathbf{P}}_{hp}[T,:]\mathbf{V}[:,i])_{hp} \tag{4}$$

where the inputs $\bar{\mathbf{P}}$ and $\mathbf{V}$ are themselves the results of prior BF16 operations. Specifically, the subscript $(\cdot)_{lp}$ here computes dot product in FP32 with the final result rounded to BF16, while $(\cdot)_{hp}$ computes entirely in FP32.

To understand how the error $\bar{\mathbf{O}}_{lp}[T,i] - \bar{\mathbf{O}}_{hp}[T,i]$ becomes systematically negative, we plot the cumulative error as the sum over token positions progresses in Fig. 6(b) and (c):

$$\bar{\mathbf{O}}_{\text{error}}(t) = \left(\sum_{t'=1}^{t} \bar{\mathbf{P}}[T,t']\mathbf{V}[t',i]\right)_{lp} - \left(\sum_{t'=1}^{t} \bar{\mathbf{P}}[T,t']\mathbf{V}[t',i]\right)_{hp}. \tag{5}$$

The figure shows that the error accumulates in significant negative steps. These steps occur at token positions $t$ where the corresponding attention probability $\bar{\mathbf{P}}[T, t]$ is exactly 1 (also observed in other token positions). This happens when the pre-softmax score $\mathbf{S}[T, t]$ is the maximum value in its row, causing $\exp(\mathbf{S}[T, t] - \max(\mathbf{S}[T, :]))$ to evaluate to $\exp(0) = 1$.

**Analysis of Biased Rounding Error** Furthermore, the bias arises from the interaction of these unit values with the distribution of the value matrix $\mathbf{V}$. As observed in Fig. 6(a), for the problematic feature dimension $i = 20$, the values of $\mathbf{V}[:, i]$ are predominantly negative. When $\bar{\mathbf{P}}[T, t] = 1$, the product $\bar{\mathbf{P}}[T, t]\mathbf{V}[t, i]$ is simply $\mathbf{V}[t, i]$, a negative BF16 number. *A systematic error then occurs when two such negative BF16 numbers are added.* In floating-point arithmetic, adding two numbers with the same sign can cause the resulting significand to overflow (e.g., $-1.\texttt{xxxx} + -1.\texttt{yyyy} = -10.\texttt{zzzz}$), requiring a right shift and an exponent increment to re-normalize. The bits shifted out of the 7-bit BF16 fraction determine the rounding direction. When adding two negative numbers, the rounding operation (e.g., round-to-nearest) can introduce a consistent bias.

To illustrate how this rounding bias occurs, consider the addition of two significands that cause an overflow, requiring a right shift for normalization. The bit that is shifted out (the rounding bit) determines the rounding direction. We show all possible additions of the last two 2-bit numbers, where the green bit represents the bit that would be shifted out (the rounding bit):

$$00 + 00 = 0\textcolor{green}{0} \qquad 00 + 01 = 0\textcolor{green}{1} \qquad 00 + 10 = 1\textcolor{green}{0} \qquad 00 + 11 = 1\textcolor{green}{1} \qquad 01 + 01 = 1\textcolor{green}{0}$$
$$01 + 10 = 1\textcolor{green}{1} \qquad 01 + 11 = 10\textcolor{green}{0} \qquad 10 + 10 = 10\textcolor{green}{0} \qquad 10 + 11 = 10\textcolor{green}{1} \qquad 11 + 11 = 11\textcolor{green}{0}$$

Because the summation is performed in FP32, the accumulation of lower-order bits from the small numbers can activate the sticky bit. Consequently, this forces a rounding up when the subsequent BF16 number is added. Therefore, the rounding bit $\textcolor{green}{1}$ indicates that rounding up is required. Because the operands $\bar{\mathbf{P}}[T, t]\mathbf{V}[t, i]$ are negative and have a large exponent (because $\bar{\mathbf{P}}[T, t] = 1$ does not make $\bar{\mathbf{P}}[T, t]\mathbf{V}[t, i]$ smaller), the error of rounding up is magnified, resulting a negative error. When the rounding bit is $\textcolor{green}{0}$, the result is rounded down, introducing a positive error. Furthermore, the positive error is smaller compared to the negative error from rounding up, as the values being added are usually very small. This asymmetry leads to the rounding error dominated by the rounding up, resulting in negative rounding error as observed in our analysis. This systematic negative bias in the computation of $\bar{\mathbf{O}}$ is the ultimate source of the training failure.

**Remark 1.** *When $\bar{\mathbf{P}}[T, t] < 1$ is less than 1 and has a mix of zero and non-zero bits in its significand's bits, its product with $\mathbf{V}[t, i]$ will also have a non-zero value in those bits. When rounding to BF16, this will not introduce biased rounding error.*

**Analysis of Rounding Error in Fig. 6** To make this concrete, we now analyze the specific BF16 number addition that causes the large negative error jump shown in Fig. 6(c). Because the summation is performed in FP32, the first BF16 value is added by some small values from other tokens before the second BF16 value is added. This can activate the sticky bit, which can force a round-up when adding the second BF16 value. For this example, we start with the FP32 representations. The first operand is the accumulated sum of previous terms, which includes a BF16 ($1\textcolor{red}{1000000}\textcolor{blue}{0001101}0000000000000000; -2.40625$) plus small residual ($\sim 0.00087$) that activate the sticky bit. The second operand is another BF16 value. Their FP32 representations are:

$$1\textcolor{red}{1000000}\textcolor{blue}{0001101}00000111000101110 \; (-2.4071154594421387)$$
$$1\textcolor{red}{1000000}\textcolor{blue}{0010011}0000000000000000 \; (-2.296875)$$

Since their exponents are identical, the addition is performed on their significands:

$$(-1.\textcolor{blue}{0011010}0000111000101110) + (-1.\textcolor{blue}{0010011}0000000000000000)$$
$$= -10.\textcolor{blue}{0101101}0000111000101110$$

The result overflows the significand's format, requiring normalization. The significand is shifted right by one bit, and the exponent is incremented:

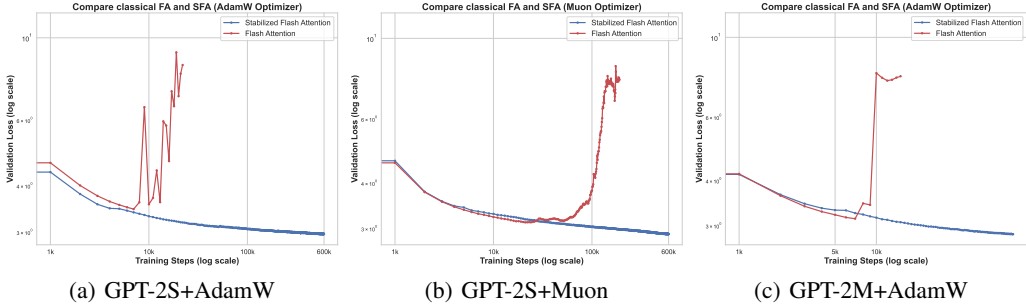

| (a) GPT-2S+AdamW | (b) GPT-2S+Muon | (c) GPT-2M+AdamW |

Figure 7: Validation losses for classical flash attention and the stablized flash attention with AdamW, Muon optimizers, and GPT-2M model.

Significand: $10.0101101 \rightarrow 1.00101101$    Exponent: $10000000 \rightarrow 10000001$

The exact result in FP32 is $1\,10000001\,00101101000011100010111$, which corresponds to $-4.703990459442139$. To store this result in BF16, it must be rounded to 7 fraction bits. The significand is $1.00101101$. The 7-bit fraction is $0010110$. Because the rounding bit is $1$ and there are nonzero bits after the rounding bit, the round-to-nearest (ties-to-even) rule rounds up (adding 1 to the last bit of the fraction):

BF16 Fraction: $0010110 + 1 = 0010111$

The final BF16 result is $1\,10000001\,0010111$, which represents $-4.71875$. This rounded value is more negative than the true sum of $-4.703990459442139$. The error introduced by this single addition is $-\mathbf{0.014759540557861328}$. When such rounding events occur systematically across many additions in the $\bar{\mathbf{P}}\mathbf{V}$ product, the errors accumulate, creating the negative bias in $\mathbf{O}$ that ultimately destabilizes training.

> **Claim 3.** *With more than one $\bar{\mathbf{P}}[T,t] = 1$ and negative $\mathbf{V}[t,i]$, the addition in $\bar{\mathbf{P}}\mathbf{V}$ can cause overflow of significand. This necessitates a right shift and a round-up of significand due to sticky bit, introducing negative rounding error in $\mathbf{O}$ and leading to positive $(\delta_{lp} - \delta_{hp})[T]$.*

## 4 EXPERIMENT: MITIGATE BIAS IN ROUNDING ERROR

Our analysis traces the failure to biased rounding errors in the $\bar{\mathbf{P}}\mathbf{V}$ computation. This occurs when multiple identical maxima in a row of the pre-softmax scores $\mathbf{S}$ cause corresponding elements in $\bar{\mathbf{P}}$ to become exactly 1. To validate our findings, we modify the softmax to detect this specific condition and adjust the normalization, ensuring all elements of $\bar{\mathbf{P}}$ are strictly less than 1. This prevents the biased rounding and restores training stability.

To prevent biased rounding, we introduce a targeted modification to the safe softmax computation. The core idea is to dynamically adjust the normalization factor $\mathbf{m}$ only when a row of the score matrix $\mathbf{S}$ contains multiple identical maximum values. This adjustment ensures that

$$\mathbf{r}_m = \text{rowmax}(\mathbf{S}), \mathbf{r}_s = \text{rowsum}(\mathbf{r}_m - \mathbf{S} \leq \epsilon)$$
$$\mathbf{m}' = \text{where}(\mathbf{r}_m > 0 \wedge \mathbf{r}_s > 1, \beta\mathbf{r}_m, \mathbf{r}_m), \beta > 1$$
$$\mathbf{m} = \text{where}(\mathbf{r}_m < 0 \wedge \mathbf{r}_s > 1, 0, \mathbf{m}')$$
$$\bar{\mathbf{P}} = \exp(\mathbf{S} - \mathbf{m})$$

the argument to the exponential function, $\mathbf{S} - \mathbf{m}$, becomes strictly negative at these maximal positions, which in turn guarantees that all elements of $\bar{\mathbf{P}} = \exp(\mathbf{S} - \mathbf{m})$ are less than 1. A naive approach, such as subtracting a small fixed constant, is insufficient as it introduces new systematic rounding errors (see Appendix C); hence, a dynamic maximum strategy is required. Our modification is presented above.

This modification prevents elements of $\bar{\mathbf{P}}$ from becoming exactly 1. We first check whether the row maximum $\mathbf{r}_m$ is attained by more than one entry by evaluating $\text{rowsum}(\mathbf{r}_m - \mathbf{S} \leq \epsilon)$, where $\epsilon$ is a

small numerical tolerance (e.g., $10^{-3}$) to account for floating-point precision. If repeated maxima are detected and $\mathbf{r}_m > 0$, the normalization factor is adjusted to $\mathbf{m} = \beta \mathbf{r}_m$ with $\beta > 1$. The new maximum in the exponent then becomes $-(\beta - 1)\mathbf{r}_m$, which is strictly negative. If instead $\mathbf{r}_m < 0$ and is repeated, we set $\mathbf{m} = 0$, which likewise ensures that the maximum exponent remains negative. In both cases, the adjustment guarantees that $\max(\mathbf{S} - \mathbf{m}) < 0$ and thus $\max(\bar{\mathbf{P}}) < 1$, preventing the conditions that lead to biased rounding.

Crucially, this modification is mathematically equivalent to standard attention in exact arithmetic, as it leverages the shift-invariance property of the softmax function ($\mathrm{softmax}(\mathbf{z}) = \mathrm{softmax}(\mathbf{z} - c)$). Our method simply chooses a different row-wise constant $c$ to ensure numerical stability. In our experiments, we set $\beta \in [2, 8]$, as smaller values risk having the result round back to 1, while larger values risk underflow. This modification is integrated into the standard flash attention tiling algorithm (line with magenta in Algorithm 3) without altering the backward pass.

To validate our approach, we pre-train the GPT-2S model for 600K steps in BF16 precision using our stabilized flash attention ($\beta = 2$) with both AdamW and Muon optimizers (Jordan et al., 2024). The AdamW configuration uses the same hyperparameters as described in Section 3.1. For the Muon setup, we employ RMS-matched Muon (Liu et al., 2025) with a learning rate of $5 \times 10^{-4}$ and weight decay of $0.01$, while retaining AdamW for embeddings, the language model head, and other one-dimensional parameters. We also extend our experiments to the larger GPT-2M model, training it for 100K steps with the AdamW optimizer under the same hyperparameters as in Section 3.1. As shown in Fig. 7, the modified flash attention successfully stabilizes training for both optimizers and larger model, preventing the loss explosion seen in the original implementation.

## 5 CONCLUSION

This paper presents the first mechanistic explanation for a notorious loss explosion in low-precision flash attention training. We pinpoint the root cause to an interplay between emergent low-rank representations and biased BF16 rounding errors. A minimal, targeted modification to flash attention validates our analysis by restoring stability. Our analytical workflow provides a blueprint for diagnosing similar numerical instabilities in other architectures, scales, and low-precision formats, paving the way for more robust and efficient large-scale model training.

**Discussion** Our findings are consistent across various hardware (NVIDIA A100, RTX 4090, Huawei Ascend 910B) and mechanistically explain empirical observations of training instability. The growth of weight spectral norms results from the accumulation of a low-rank error matrix in the gradients. We also clarify the role of attention sinks: by attracting high attention scores, they are more likely to produce attention probabilities of 1, which triggers the biased rounding error in the $\bar{\mathbf{P}}\mathbf{V}$ computation. This provides a direct numerical link between the architectural behavior of sinks and the arithmetic instability that derails training. Finally, our analysis provides a compelling explanation for the effectiveness of established stabilization techniques. We posit that a root cause of training failure is the structural similarity within the error matrices, $(\mathbf{PK})[T]^\top \mathbf{X}[T]$, which creates a pathway for rounding errors to accumulate systematically. Techniques like QK normalization and Gated Attention disrupt this underlying structure. In doing so, they ensure that even when rounding errors are present, they lack the coherence to compound, thereby averting the instabilities observed during training.

**Limitations** Our analysis focuses on a specific failure case in a GPT-2 model. The generalizability of our findings to other architectures, larger scales, or different low-precision formats like FP8 requires further investigation. Additionally, our proposed mitigation is tailored to the specific rounding error identified and may not address other sources of numerical instability.

**Future Work** Our work provides a foundation for exploring numerical stability in low-precision training. Our analytical framework, i.e., locating the source of error, finding similar update directions, and tracing back to root causes, can be applied to other architectures and precision formats. Future work could extend this analysis to FP8 training, larger models, and different architectures. Additionally, developing automated tools to detect and mitigate such numerical instabilities during training would be valuable for the community.

ACKNOWLEDGMENT

This work is supported by Beijing Science and Technology Program (No.Z251100008125003).

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

# A    RELATED WORK

## A.1    MIXED-PRECISION BF16 TRAINING.

Contemporary large language model (LLM) pretraining almost universally employs mixed-precision arithmetic. Early efforts by Micikevicius et al. (2017) demonstrated that FP16 training—using an FP32 master copy of weights and fixed loss scaling—could match FP32 accuracy for many models. However, the narrow exponent range of FP16 often causes many gradients to underflow, necessitating careful tuning. The bfloat16 (BF16) format, with an 8-bit exponent and 7-bit mantissa, retains the wide dynamic range of FP32 while halving storage cost. Kalamkar et al. (2019) showed that BF16 achieves convergence parity with FP32 on large models without specialized tuning. Since then, BF16 has become the default 16-bit format for many large-scale training frameworks, with native support in PyTorch and TensorFlow (Wang & Kanwar, 2019).

BF16 mixed precision has enabled training of landmark LLMs at unprecedented scales, including GPT-3 (175B) (Brown et al., 2020), Google PaLM (540B) (Chowdhery et al., 2023), DeepMind Gopher (280B) (Rae et al., 2021), Chinchilla (70B) (Hoffmann et al., 2022), and Meta's LLaMA family (7B-65B) (Touvron et al., 2023). To handle the massive memory footprint, parallel training frameworks such as Megatron and DeepSpeed integrate BF16 training with techniques like the Zero Redundancy Optimizer (ZeRO) (Rajbhandari et al., 2020).

Despite its advantages, BF16 training can still exhibit instabilities. Empirical studies show that FP16 is highly unstable even with loss scaling, whereas BF16 eliminates most precision-related tuning and failures Wang & Kanwar (2019). However, Lee et al. (2024) report that roughly 10% of GPT-2 pretraining runs diverged under pure BF16, compared to 0% under TF32. This suggests that while BF16 substantially improves stability, complementary stabilization techniques remain necessary at scale.

## A.2    STABILIZING LOW-PRECISION TRAINING

**Gradient Scaling.**    Early work by Micikevicius et al. (2017) introduced FP16 mixed-precision training, where weights, activations, and gradients are stored in half-precision while maintaining a master FP32 copy. They also proposed *loss scaling* to prevent FP16 underflows. Even with loss scaling, some underflow can still occur in deep networks. To address this, Zhao et al. (2021) introduced *gradient scaling*, which dynamically computes per-layer scaling factors to avoid both underflow and overflow.

**Ultra-Low-Precision (FP8/INT8) Training.**    To further reduce cost, recent works explore FP8 or INT8 precision for training and inference. However, naive FP8 training tends to diverge. Lee et al. (2024) note that the direct application of FP8 to LLM training is unstable without additional stabilization techniques. To address this, Perez et al. (2023) propose dynamically adjusted per-tensor scaling factors for FP8 matrix multiplications. Using this scheme, they successfully train GPT- and LLaMA-style models up to 70B parameters entirely in FP8. Similarly, Peng et al. (2023) introduce FP8-LM, a framework that progressively applies FP8 to gradients, optimizer states, and distributed communication, achieving a 39% memory reduction and 75% speedup compared to BF16. Balança et al. (2024) present SCALIFY, which propagates scale factors throughout the computation graph to ensure stable FP8 operations without manual tuning. These approaches collectively demonstrate that careful scaling management enables FP8 or INT8 training to match BF16 performance while reducing memory and compute requirements.

**Optimizer and Gradient Stabilization.**    Optimizer algorithms play a critical role in training stability. Molybog et al. (2023) theoretically analyze Adam and show that catastrophic divergence often arises when the update direction becomes uncorrelated with the true descent direction in large-scale models. To address gradient instability, Huang et al. (2025) propose SPAM (Spike-Aware Adam with Momentum Reset), which detects and mitigates rare but severe "gradient spikes" by resetting momentum and applying spike-aware clipping. In parallel, Wortsman et al. (2023) investigate loss spikes in vision-language models and show that AdamW often underestimates the second moment before spikes occur. They propose a hybrid AdamW-AdaFactor optimizer that adaptively corrects

second-moment underestimation, outperforming gradient clipping alone. These methods highlight how optimizer modifications directly mitigate divergence in low-precision regimes.

**Activation and Architectural Techniques.**   The choice of activation functions and initialization strategies also impacts stability. Fishman et al. (2024) observe that the SwiGLU activation amplifies outliers during long FP8 training runs. They introduce *Smooth-SwiGLU*, a modified activation that prevents outlier amplification, enabling stable trillion-token FP8 training. In the vision-language domain, Wortsman et al. (2023) show that "layer-scale zero" initialization and carefully designed low-precision linear layers (e.g., SwitchBack) further improve stability in int8 training.

### A.3   FLASH ATTENTION

Flash Attention (Dao et al., 2022) is an I/O-aware exact attention algorithm designed to overcome the memory bottleneck of standard self-attention, which scales quadratically with sequence length $N$. Standard attention requires materializing the full $N \times N$ attention score matrix, leading to $\mathcal{O}(N^2)$ memory usage that becomes prohibitive for long sequences. Flash Attention mitigates this by employing a tiling and kernel fusion strategy. The query, key, and value matrices are partitioned into smaller blocks, which are iteratively loaded from slow high-bandwidth memory (HBM) into fast on-chip SRAM.

Within the SRAM, the core computation is performed. A key innovation is the use of an online softmax algorithm. Instead of computing the full score matrix before normalization, Flash Attention maintains running statistics (the maximum score and the sum of exponentials) as it processes blocks of keys and values. This allows it to compute the correctly normalized output block-by-block without ever storing the entire intermediate attention matrix. By fusing the matrix multiplications and softmax operations into a single GPU kernel, it drastically reduces the number of memory read/write operations to HBM, which is the primary performance bottleneck. This I/O-aware design reduces the memory complexity from $\mathcal{O}(N^2)$ to $\mathcal{O}(N)$ and significantly accelerates computation. Flash Attention 2 (Dao, 2024) further optimized this approach by improving work partitioning among GPU thread blocks and warps and reducing non-matmul computation overhead, achieving near-optimal performance and making it an indispensable component for training and inferring long-context LLMs. In the pseudocode below, we absorb the standard attention scaling factor $\alpha = 1/\sqrt{d}$ into $\mathbf{Q}$ for notational simplicity. See Algorithm 1 and Algorithm 2 for pseudocode of Flash Attention 2's forward and backward passes.

## B   BF16 ADDITION

The bfloat16 (Brain Floating-Point) format is a 16-bit floating-point representation widely used in deep learning for its balance between computational efficiency and numerical range. It consists of 1 sign bit, 8 exponent bits, and 7 fraction (or mantissa) bits. This structure gives bfloat16 the same dynamic range as the 32-bit single-precision format (FP32) but with significantly less precision.

The addition of two bfloat16 numbers, say $a$ and $b$, follows the standard procedure for floating-point arithmetic:

1. **Exponent Alignment:** The exponents of the two numbers are compared. The number with the smaller exponent has its significand (the combination of the implicit leading bit and the fraction) shifted to the right until its exponent matches the larger one. Each right shift increases the exponent by one. Bits shifted past the available precision are lost, which is an initial source of error.

2. **Significand Addition:** The aligned significands are added together. The sign of the result is determined by the signs and magnitudes of the operands.

3. **Normalization:** The result is normalized to ensure it conforms to the 1.xxxx... $\times 2^e$ format. If the addition resulted in an overflow (e.g., 10.xxxx...), the significand is shifted right and the exponent is incremented. If it resulted in cancellation (e.g., 0.00xx...), the significand is shifted left and the exponent is decremented until the leading bit is 1.

4. **Rounding:** The resulting significand, which may have more than 7 fraction bits after normalization, must be rounded. The standard mode is "round to nearest, ties to even". This means if the

---

**Algorithm 1** Flash Attention: Forward Pass

---

**Require:** Matrices $\mathbf{Q}, \mathbf{K}, \mathbf{V} \in \mathbb{R}^{N \times d}$, block sizes $B_c, B_r$.

1: Divide $\mathbf{Q}$ into $T_r = \left\lceil \frac{N}{B_r} \right\rceil$ blocks $\mathbf{Q}_1, \ldots, \mathbf{Q}_{T_r}$ of size $B_r \times d$ each, and divide $\mathbf{K}, \mathbf{V}$ in to $T_c = \left\lceil \frac{N}{B_c} \right\rceil$ blocks $\mathbf{K}_1, \ldots, \mathbf{K}_{T_c}$ and $\mathbf{V}_1, \ldots, \mathbf{V}_{T_c}$, of size $B_c \times d$ each.

2: Divide the output $\mathbf{O} \in \mathbb{R}^{N \times d}$ into $T_r$ blocks $\mathbf{O}_1, \ldots, \mathbf{O}_{T_r}$ of size $B_r \times d$ each, and divide the logsumexp $\mathbf{L}$ into $T_r$ blocks $\mathbf{L}_1, \ldots, \mathbf{L}_{T_r}$ of size $B_r$ each.

3: **for** $1 \leq i \leq T_r$ **do**

4:     Initialize $\mathbf{O}_i^{(0)} = (0)_{B_r \times d} \in \mathbb{R}^{B_r \times d}, \boldsymbol{\ell}_i^{(0)} = (0)_{B_r} \in \mathbb{R}^{B_r}, \mathbf{m}_i^{(0)} = (-\infty)_{B_r} \in \mathbb{R}^{B_r}$.

5:     **for** $1 \leq j \leq T_c$ **do**

6:         Compute $\mathbf{S}_i^{(j)} = \mathbf{Q}_i \mathbf{K}_j^T \in \mathbb{R}^{B_r \times B_c}$.

7:         Compute $\mathbf{m}_i^{(j)} = \max(\mathbf{m}_i^{(j-1)}, \mathrm{rowmax}(\mathbf{S}_i^{(j)})) \in \mathbb{R}^{B_r}, \bar{\mathbf{P}}_i^{(j)} = \exp(\mathbf{S}_i^{(j)} - \mathbf{m}_i^{(j)}) \in \mathbb{R}^{B_r \times B_c}$ (pointwise), $\boldsymbol{\ell}_i^{(j)} = e^{\mathbf{m}_i^{(j-1)} - \mathbf{m}_i^{(j)}} \boldsymbol{\ell}_i^{(j-1)} + \mathrm{rowsum}(\bar{\mathbf{P}}_i^{(j)}) \in \mathbb{R}^{B_r}$.

8:         Compute $\mathbf{O}_i^{(j)} = \mathrm{diag}(e^{\mathbf{m}_i^{(j-1)} - \mathbf{m}_i^{(j)}})\mathbf{O}_i^{(j-1)} + \bar{\mathbf{P}}_i^{(j)} \mathbf{V}_j$.

9:     **end for**

10:    Compute $\mathbf{O}_i = \mathrm{diag}(\boldsymbol{\ell}_i^{(T_c)})^{-1} \mathbf{O}_i^{(T_c)}$.

11:    Compute $\mathbf{L}_i = \mathbf{m}_i^{(T_c)} + \log(\boldsymbol{\ell}_i^{(T_c)})$.

12:    Write $\mathbf{O}_i$ as the $i$-th block of $\mathbf{O}$.

13:    Write $\mathbf{L}_i$ as the $i$-th block of $\mathbf{L}$.

14: **end for**

15: Return the output $\mathbf{O}$ and the logsumexp $\mathbf{L}$.

---

**Algorithm 2** Flash Attention: Backward Pass

---

**Require:** Matrices $\mathbf{Q}, \mathbf{K}, \mathbf{V}, \mathbf{O}, d\mathbf{O} \in \mathbb{R}^{N \times d}$, vector $\mathbf{L} \in \mathbb{R}^N$, block sizes $B_c, B_r$.

1: Divide $\mathbf{Q}$ into $T_r = \left\lceil \frac{N}{B_r} \right\rceil$ blocks $\mathbf{Q}_1, \ldots, \mathbf{Q}_{T_r}$ of size $B_r \times d$ each, and divide $\mathbf{K}, \mathbf{V}$ in to $T_c = \left\lceil \frac{N}{B_c} \right\rceil$ blocks $\mathbf{K}_1, \ldots, \mathbf{K}_{T_c}$ and $\mathbf{V}_1, \ldots, \mathbf{V}_{T_c}$, of size $B_c \times d$ each.

2: Divide $\mathbf{O}$ into $T_r$ blocks $\mathbf{O}_1, \ldots, \mathbf{O}_{T_r}$ of size $B_r \times d$ each, divide $d\mathbf{O}$ into $T_r$ blocks $d\mathbf{O}_1, \ldots, d\mathbf{O}_{T_r}$ of size $B_r \times d$ each, and divide $\mathbf{L}$ into $T_r$ blocks $\mathbf{L}_1, \ldots, \mathbf{L}_{T_r}$ of size $B_r$ each.

3: Initialize $d\mathbf{Q} = (0)_{N \times d}$ and divide it into $T_r$ blocks $d\mathbf{Q}_1, \ldots, d\mathbf{Q}_{T_r}$ of size $B_r \times d$ each. Divide $d\mathbf{K}, d\mathbf{V} \in \mathbb{R}^{N \times d}$ in to $T_c$ blocks $d\mathbf{K}_1, \ldots, d\mathbf{K}_{T_c}$ and $d\mathbf{V}_1, \ldots, d\mathbf{V}_{T_c}$, of size $B_c \times d$ each.

4: Compute $\boldsymbol{\delta} = \mathrm{rowsum}(d\mathbf{O} \circ \mathbf{O}) \in \mathbb{R}^N$ (pointwise multiply), and divide it into $T_r$ blocks $\boldsymbol{\delta}_1, \ldots, \boldsymbol{\delta}_{T_r}$ of size $B_r$ each.

5: **for** $1 \leq j \leq T_c$ **do**

6:     Initialize $d\mathbf{K}_j = (0)_{B_c \times d}, d\mathbf{V}_j = (0)_{B_c \times d}$.

7:     **for** $1 \leq i \leq T_r$ **do**

8:         Compute $\mathbf{S}_i^{(j)} = \mathbf{Q}_i \mathbf{K}_j^T \in \mathbb{R}^{B_r \times B_c}$.

9:         Compute $\mathbf{P}_i^{(j)} = \exp(\mathbf{S}_i^{(j)} - \mathbf{L}_i) \in \mathbb{R}^{B_r \times B_c}$.

10:       Compute $d\mathbf{V}_j \leftarrow d\mathbf{V}_j + (\mathbf{P}_i^{(j)})^\top d\mathbf{O}_i \in \mathbb{R}^{B_c \times d}$.

11:       Compute $d\mathbf{P}_i^{(j)} = d\mathbf{O}_i \mathbf{V}_j^\top \in \mathbb{R}^{B_r \times B_c}$.

12:       Compute $d\mathbf{S}_i^{(j)} = \mathbf{P}_i^{(j)} \circ (d\mathbf{P}_i^{(j)} - \boldsymbol{\delta}_i) \in \mathbb{R}^{B_r \times B_c}$.

13:       Update $d\mathbf{Q}_i \leftarrow d\mathbf{Q}_i + d\mathbf{S}_i^{(j)} \mathbf{K}_j \in \mathbb{R}^{B_r \times d}$.

14:       Compute $d\mathbf{K}_j \leftarrow d\mathbf{K}_j + {d\mathbf{S}_i^{(j)}}^\top \mathbf{Q}_i \in \mathbb{R}^{B_c \times d}$.

15:     **end for**

16: **end for**

17: Return $d\mathbf{Q}, d\mathbf{K}, d\mathbf{V}$.

---

truncated part is greater than half the value of the last storable bit (LSB), the number is rounded up. If it is less, it is rounded down. If it is exactly half, it is rounded to the nearest value with an even LSB.

The key source of numerical error comes from steps 1 and 4. During exponent alignment, precision is lost from the smaller-magnitude number. After addition, the result must be rounded back to the 7-bit fraction, which introduces another rounding error. While "round to nearest, ties to even" is designed to be unbiased for random data, a sequence of additions on data with a specific distribution (e.g., mostly negative numbers being added together) can lead to a *biased rounding error*, where the accumulated error consistently pushes the result in one direction. This accumulation of biased error is a critical factor in the training failure observed in low-precision settings.

## C  DESIGN CONSIDERATIONS FOR MITIGATING BIASED ROUNDING ERROR IN FLASH ATTENTION

**Use Dynamic Maximum Value Rather Than Fixed Offset**    A fixed offset would cause the computed values of $\bar{\mathbf{P}}$ to be consistently rounded in one direction during BF16 conversion, introducing a fixed error. Since the elements of $\mathbf{V}$ often share the same sign, this fixed rounding error in $\bar{\mathbf{P}}$ does not average out to zero when computing $\bar{\mathbf{P}}\mathbf{V}$. This leads to a biased error in the output $\mathbf{O}$, which in turn creates a biased term $\boldsymbol{\delta}$, reintroducing the very failure we aim to solve.

**Dynamic Maximum is Applied Conditionally**    Our modification is applied conditionally—only when a row contains multiple identical maximum values—to avoid introducing new numerical instabilities. An unconditional adjustment is not a better option. For example, if a row has a single, very large positive maximum value $\mathbf{r}_m$, applying our rule would mean calculating $\exp(\mathbf{S} - \beta\mathbf{r}_m)$. The largest term in the exponent would become $-(\beta - 1)\mathbf{r}_m$, and $\exp(-(\beta - 1)\mathbf{r}_m)$ could underflow to zero. This would cause the normalization factor to become zero, leading to a division-by-zero error when computing the output $\mathbf{O}$. By applying the modification only in the specific case that causes biased rounding, we preserve the numerical stability of the standard online softmax in all other scenarios.

**Explanation on Dealing Negative Repeated Row Maximum**    We also explore alternative stabilization methods for the negative, repeated row maximum ($\mathbf{r}_m < 0$). One approach involves setting the normalization factor $\mathbf{m} = \gamma\mathbf{r}_m$ for some $\gamma \in (0, 1)$. This makes the new maximum value in the exponent $(1 - \gamma)\mathbf{r}_m$. However, we observe that if $\gamma$ is close to 1, this new maximum approaches zero. In low-precision arithmetic, $\exp((1 - \gamma)\mathbf{r}_m)$ can round to exactly 1, reintroducing the very failure we aim to prevent. Consequently, we found that setting $\gamma = 0$ (i.e., $\mathbf{m} = 0$) is a robust choice, as it ensures the maximum value in the exponent remains sufficiently negative.

## D  SIMILAR PATTERNS DISCOVERED IN LLAMA-3.1-8B

To see if the phenomena that leads to the failure case in GPT-2 also exists in other models, we present $(\mathbf{PK})[T]^\top\mathbf{X}[T]$ and the attention score in Llama-3.1-8B which shows the similar patterns we discovered in GPT-2. This section is not meant to say Llama 3.1-8B will definitely fail in BF16 training, but rather to show that the phenomena we discovered is not unique to GPT-2.

### D.1  STRUCTURALLY SIMILAR $(\mathbf{PK})[T]^\top\mathbf{X}[T]$

Given an input "The quick brown fox jumps over the lazy dog", we visualize $(\mathbf{PK})[T]^\top\mathbf{X}[T]$ for attention head 13 in layer 1 of Llama-3.1-8B. We visualize the representations and similar columns in Fig. 8.

### D.2  MULTIPLE MAXIMUM IN $\bar{\mathbf{P}}$

We also observe multiple maximum values in the attention scores of Llama-3.1-8B. We visualize some examples in Fig. 9. This experiment shows the attention sink (Xiao et al., 2023) behavior as well, where the first token has high attention scores.

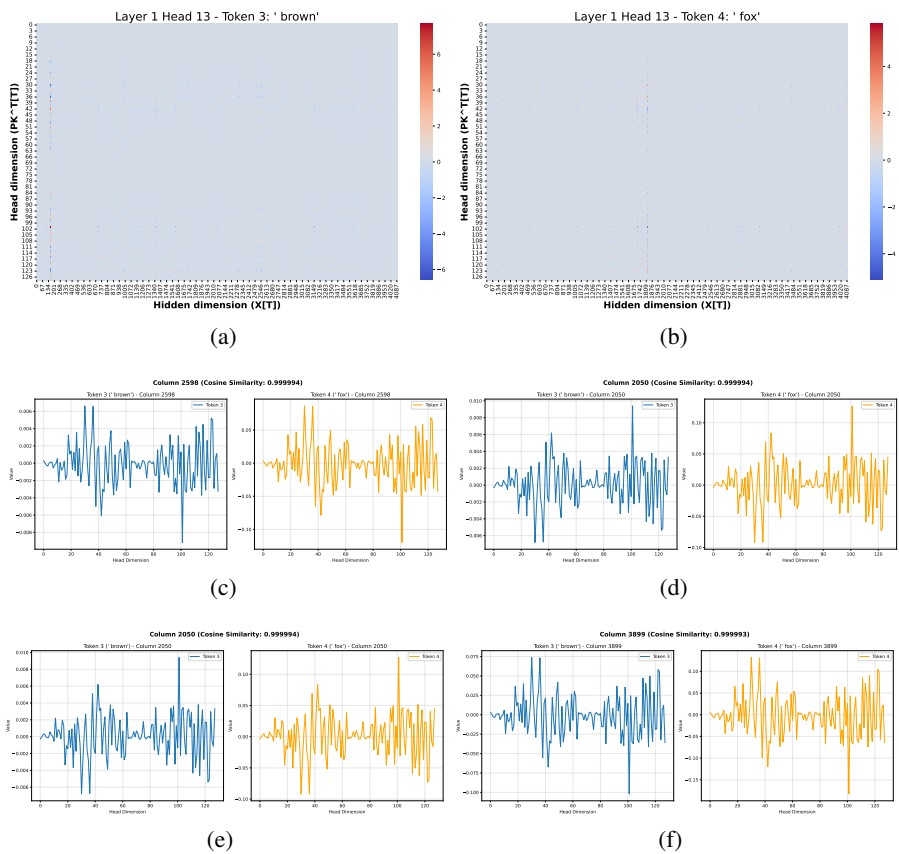

Figure 8: Similar $(\mathbf{PK})[T]^\top \mathbf{X}[T]$ discovered at different tokens in Llama-3.1-8B. (c)-(f) show the similar columns.

# E  OTHER DETAILS

## E.1  RELATION BETWEEN MULTIPLE MAXIMA AND LOSS

To investigate the connection between the multiple maxima phenomenon and training instability, we plot the frequency of multiple maxima occurrences against the training loss in Fig. 13. The visualization reveals a clear temporal pattern: the number of multiple maxima begins to increase before the loss explodes (see 7000 step). This strong correlation suggests that the emergence of multiple maxima is a leading indicator of the impending training failure.

## E.2  DERIVING EQUATION 1

The following equations are prepared to derive equation 1 in the main text.

$$dQ = dSK \tag{6}$$
$$dS = \alpha \mathbf{P} \circ (d\mathbf{P} - \boldsymbol{\delta}) \tag{7}$$

Here is the detailed derivation.

We begin with the goal of deriving an expression for the difference $d\mathbf{Q}_{hp} - d\mathbf{Q}_{lp}$.

1. Apply the definition of $d\mathbf{Q}$. We start with the left-hand side of the equation and substitute the definition from equation 6 for both the high-precision (hp) and low-precision (lp) terms.

$$d\mathbf{Q}_{hp} - d\mathbf{Q}_{lp} = (d\mathbf{S}_{hp}\mathbf{K}) - (d\mathbf{S}_{lp}\mathbf{K})$$

Using the right distributive property of matrix multiplication, we can factor out the matrix $\mathbf{K}$.

$$d\mathbf{Q}_{hp} - d\mathbf{Q}_{lp} = (d\mathbf{S}_{hp} - d\mathbf{S}_{lp})\,\mathbf{K} \tag{8}$$

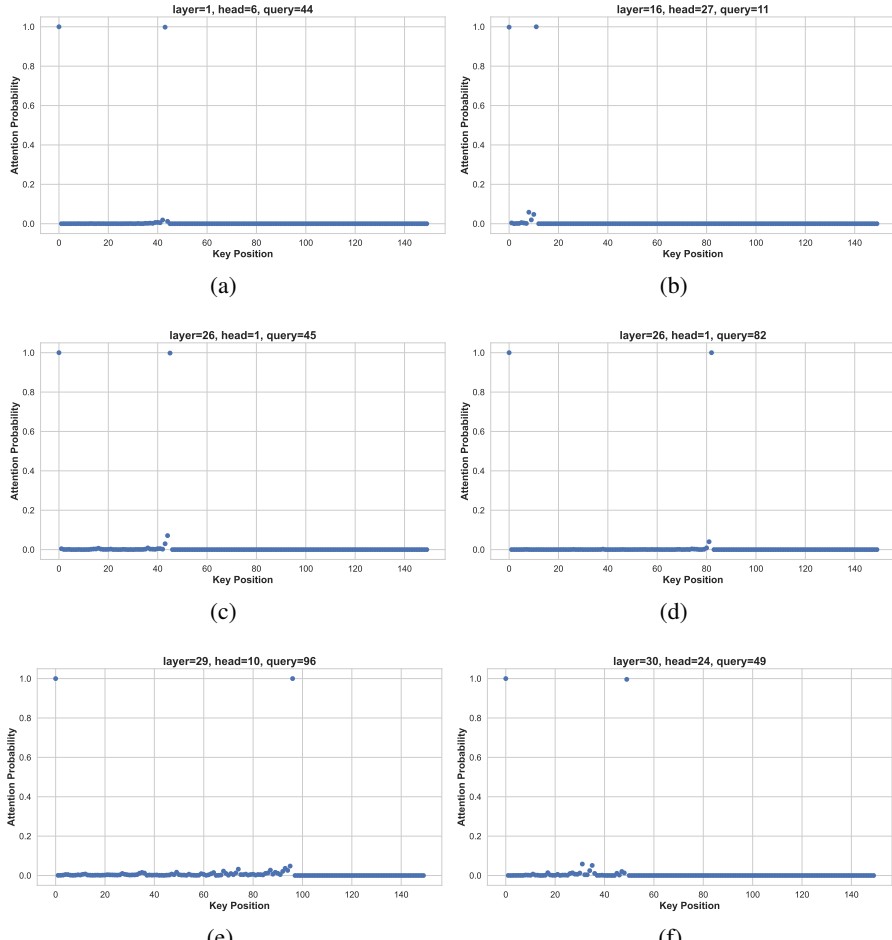

Figure 9: Two maxima observed in Llama-3.1-8B. The first token has maxima due to the attention sink.

2. Substitute the definition of $d\mathbf{S}$. Next, we substitute the expressions for $d\mathbf{S}_{hp}$ and $d\mathbf{S}_{lp}$ using the definition from equation 7.

$$d\mathbf{S}_{hp} = \alpha\mathbf{P} \circ (d\mathbf{P} - \boldsymbol{\delta}_{hp})$$
$$d\mathbf{S}_{lp} = \alpha\mathbf{P} \circ (d\mathbf{P} - \boldsymbol{\delta}_{lp})$$

Plugging these into our expression equation 8:

$$d\mathbf{Q}_{hp} - d\mathbf{Q}_{lp} = [(\alpha\mathbf{P} \circ (d\mathbf{P} - \boldsymbol{\delta}_{hp})) - (\alpha\mathbf{P} \circ (d\mathbf{P} - \boldsymbol{\delta}_{lp}))]\mathbf{K} \tag{9}$$

3. Simplify using properties of the Hadamard product. The Hadamard product ($\circ$) is distributive over matrix addition and subtraction, and we can factor out the common scalar $\alpha$ and matrix $\mathbf{P}$.

$$d\mathbf{Q}_{hp} - d\mathbf{Q}_{lp} = [\alpha\,(\mathbf{P} \circ (d\mathbf{P} - \boldsymbol{\delta}_{hp}) - \mathbf{P} \circ (d\mathbf{P} - \boldsymbol{\delta}_{lp}))]\mathbf{K}$$
$$= [\alpha\mathbf{P} \circ ((d\mathbf{P} - \boldsymbol{\delta}_{hp}) - (d\mathbf{P} - \boldsymbol{\delta}_{lp}))]\mathbf{K}$$

Now, we simplify the expression inside the parentheses. The $d\mathbf{P}$ terms cancel each other out.

$$d\mathbf{Q}_{hp} - d\mathbf{Q}_{lp} = [\alpha\mathbf{P} \circ (d\mathbf{P} - \boldsymbol{\delta}_{hp} - d\mathbf{P} + \boldsymbol{\delta}_{lp})]\mathbf{K}$$
$$= [\alpha\mathbf{P} \circ (\boldsymbol{\delta}_{lp} - \boldsymbol{\delta}_{hp})]\mathbf{K}$$

4. Convert the Hadamard product to matrix multiplication. The final step involves rewriting the row-wise scaling operation, represented by the Hadamard product with broadcasting, as a standard matrix multiplication. Let the vector difference be $\mathbf{v} = \boldsymbol{\delta}_{lp} - \boldsymbol{\delta}_{hp}$. The expression $\mathbf{P} \circ (\boldsymbol{\delta}_{lp} - \boldsymbol{\delta}_{hp})$

---

**Algorithm 3** Stabilized Flash Attention by Mitigating Biased Rounding Error: Forward Pass

---

**Require:** Matrices $\mathbf{Q}, \mathbf{K}, \mathbf{V} \in \mathbb{R}^{N \times d}$, block sizes $B_c, B_r, \beta > 1, \epsilon > 0$.

1: Divide $\mathbf{Q}$ into $T_r = \left\lceil \frac{N}{B_r} \right\rceil$ blocks $\mathbf{Q}_1, \ldots, \mathbf{Q}_{T_r}$ of size $B_r \times d$ each, and divide $\mathbf{K}, \mathbf{V}$ in to $T_c = \left\lceil \frac{N}{B_c} \right\rceil$ blocks $\mathbf{K}_1, \ldots, \mathbf{K}_{T_c}$ and $\mathbf{V}_1, \ldots, \mathbf{V}_{T_c}$, of size $B_c \times d$ each.

2: Divide the output $\mathbf{O} \in \mathbb{R}^{N \times d}$ into $T_r$ blocks $\mathbf{O}_1, \ldots, \mathbf{O}_{T_r}$ of size $B_r \times d$ each, and divide the logsumexp $\mathbf{L}$ into $T_r$ blocks $\mathbf{L}_1, \ldots, \mathbf{L}_{T_r}$ of size $B_r$ each.

3: **for** $1 \leq i \leq T_r$ **do**

4:     Initialize $\mathbf{O}_i^{(0)} = (0)_{B_r \times d} \in \mathbb{R}^{B_r \times d}, \boldsymbol{\ell}_i^{(0)} = (0)_{B_r} \in \mathbb{R}^{B_r}, \mathbf{m}_i^{(0)} = (-\infty)_{B_r} \in \mathbb{R}^{B_r}$.

5:     **for** $1 \leq j \leq T_c$ **do**

6:         Compute $\mathbf{S}_i^{(j)} = \mathbf{Q}_i \mathbf{K}_j^T \in \mathbb{R}^{B_r \times B_c}$.

7:         $\mathbf{r}_m = \text{rowmax}(\mathbf{S}_i^{(j)}), \mathbf{r}_s = \text{rowsum}(\mathbf{r}_m - \mathbf{S}_i^{(j)} \leq \epsilon)$

8:         $\mathbf{m}_i^{(j)\prime} = \text{where}(\mathbf{r}_m > 0 \wedge \mathbf{r}_s > 1, \beta \mathbf{r}_m, \mathbf{r}_m)$

9:         $\mathbf{m}_i^{(j)\prime} = \text{where}(\mathbf{r}_m < 0 \wedge \mathbf{r}_s > 1, 0, \mathbf{m}_i^{(j)\prime})$

10:        Compute $\mathbf{m}_i^{(j)} = \max(\mathbf{m}_i^{(j-1)}, \mathbf{m}_i^{(j)\prime}) \in \mathbb{R}^{B_r}, \bar{\mathbf{P}}_i^{(j)} = \exp(\mathbf{S}_i^{(j)} - \mathbf{m}_i^{(j)}) \in \mathbb{R}^{B_r \times B_c}$ (pointwise), $\boldsymbol{\ell}_i^{(j)} = e^{\mathbf{m}_i^{(j-1)} - \mathbf{m}_i^{(j)}} \boldsymbol{\ell}_i^{(j-1)} + \text{rowsum}(\bar{\mathbf{P}}_i^{(j)}) \in \mathbb{R}^{B_r}$.

11:        Compute $\mathbf{O}_i^{(j)} = \text{diag}(e^{\mathbf{m}_i^{(j-1)} - \mathbf{m}_i^{(j)}}) \mathbf{O}_i^{(j-1)} + \bar{\mathbf{P}}_i^{(j)} \mathbf{V}_j$.

12:     **end for**

13:     Compute $\mathbf{O}_i = \text{diag}(\boldsymbol{\ell}_i^{(T_c)})^{-1} \mathbf{O}_i^{(T_c)}$.

14:     Compute $\mathbf{L}_i = \mathbf{m}_i^{(T_c)} + \log(\boldsymbol{\ell}_i^{(T_c)})$.

15:     Write $\mathbf{O}_i$ as the $i$-th block of $\mathbf{O}$.

16:     Write $\mathbf{L}_i$ as the $i$-th block of $\mathbf{L}$.

17: **end for**

18: Return the output $\mathbf{O}$ and the logsumexp $\mathbf{L}$.

---

means that each row of matrix $\mathbf{P}$ is element-wise multiplied by the vector $\mathbf{v}$. Specifically, the $i$-th row of $\mathbf{P}$ is scaled by the $i$-th element of $\mathbf{v}$.

Let's examine this at the element level. The $(i, j)$-th element of the resulting matrix is:

$$(\mathbf{P} \circ \mathbf{v})_{ij} = \mathbf{P}_{ij} \cdot \mathbf{v}_i$$

This is precisely the result of pre-multiplying the matrix $\mathbf{P}$ by a diagonal matrix whose diagonal entries are the elements of the vector $\mathbf{v}$. Let $\mathbf{D} = \text{diag}(\mathbf{v}) = \text{diag}(\boldsymbol{\delta}_{lp} - \boldsymbol{\delta}_{hp})$. The $(i, j)$-th element of the product $\mathbf{DP}$ is:

$$(\mathbf{DP})_{ij} = \sum_k \mathbf{D}_{ik} \mathbf{P}_{kj}$$

Since $\mathbf{D}$ is a diagonal matrix, $\mathbf{D}_{ik}$ is non-zero only when $k = i$, where $\mathbf{D}_{ii} = \mathbf{v}_i$. Thus, the sum simplifies to:

$$(\mathbf{DP})_{ij} = \mathbf{D}_{ii} \mathbf{P}_{ij} = \mathbf{v}_i \cdot \mathbf{P}_{ij}$$

This confirms that $\mathbf{P} \circ \mathbf{v} = \text{diag}(\mathbf{v})\mathbf{P}$. Applying this identity to our expression equation 10:

$$d\mathbf{Q}_{hp} - d\mathbf{Q}_{lp} = [\alpha \, \text{diag}(\boldsymbol{\delta}_{lp} - \boldsymbol{\delta}_{hp})\mathbf{P}] \, \mathbf{K}$$

Finally, using the associativity of matrix multiplication, we can regroup the terms to arrive at the final equation.

$$d\mathbf{Q}_{hp} - d\mathbf{Q}_{lp} = \alpha \, \text{diag}(\boldsymbol{\delta}_{lp} - \boldsymbol{\delta}_{hp})(\mathbf{PK}) \tag{10}$$

This completes the detailed derivation from the initial definitions to the final expression.

## F   SUGGESTIONS FOR HANDLING FAILURE IN LOW-PRECISION TRAINING

Our analysis provides a blueprint for diagnosing numerical instabilities that may arise in other low-precision formats, such as FP8. The analytical workflow presented in this paper is possible to be generalized to other settings. The key steps are:

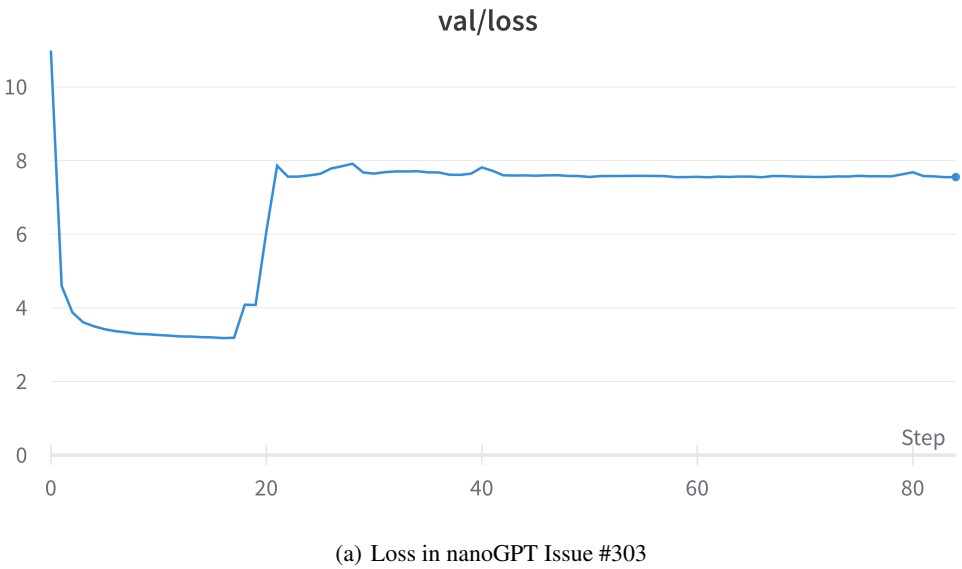

(a) Loss in nanoGPT Issue #303

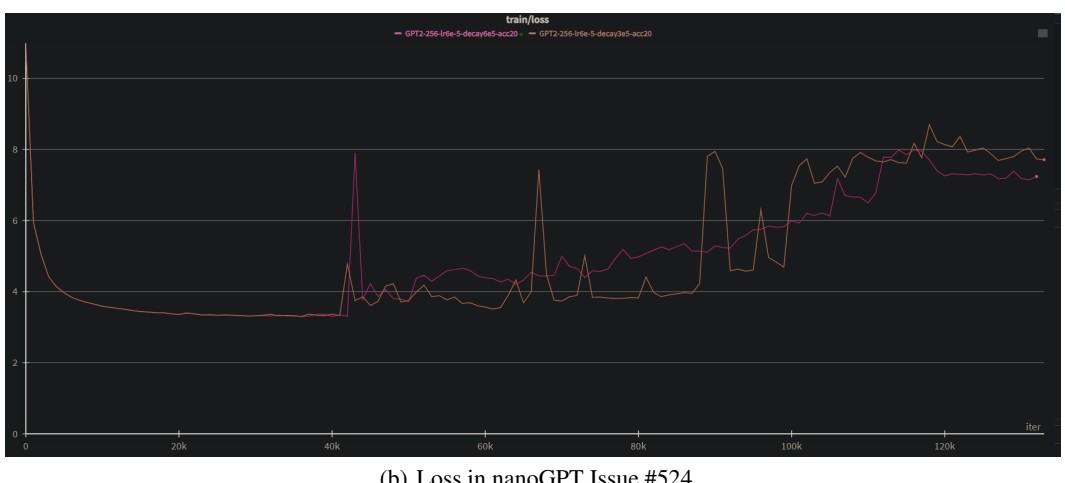

(b) Loss in nanoGPT Issue #524

Figure 10: Loss curves of two independent runs of GPT-2 training with flash attention in BF16 reported in the Github issue of nanoGPT.

1. **Isolate the Source of Error:** Systematically narrow down the failure to a specific operation or module through targeted high-precision substitutions.

2. **Identify Error Accumulation Mechanisms:** Analyze the gradient error to find structural patterns, such as the emergence of similar update directions that allow errors to accumulate rather than cancel out.

3. **Trace to Root Arithmetic Cause:** Examine the low-level arithmetic of the specific format to find the origin of any systematic bias.

By following this methodology, researchers and practitioners in this field can move from observing a high-level training failure to understanding its root cause in the underlying hardware arithmetic, enabling the development of principled solutions.

## THE USE OF LARGE LANGUAGE MODELS (LLMS)

We use LLMs for various purposes, including: polishing the language of the paper and assistant code implementation. We carefully review and edit all content generated by LLMs to ensure accuracy and appropriateness.

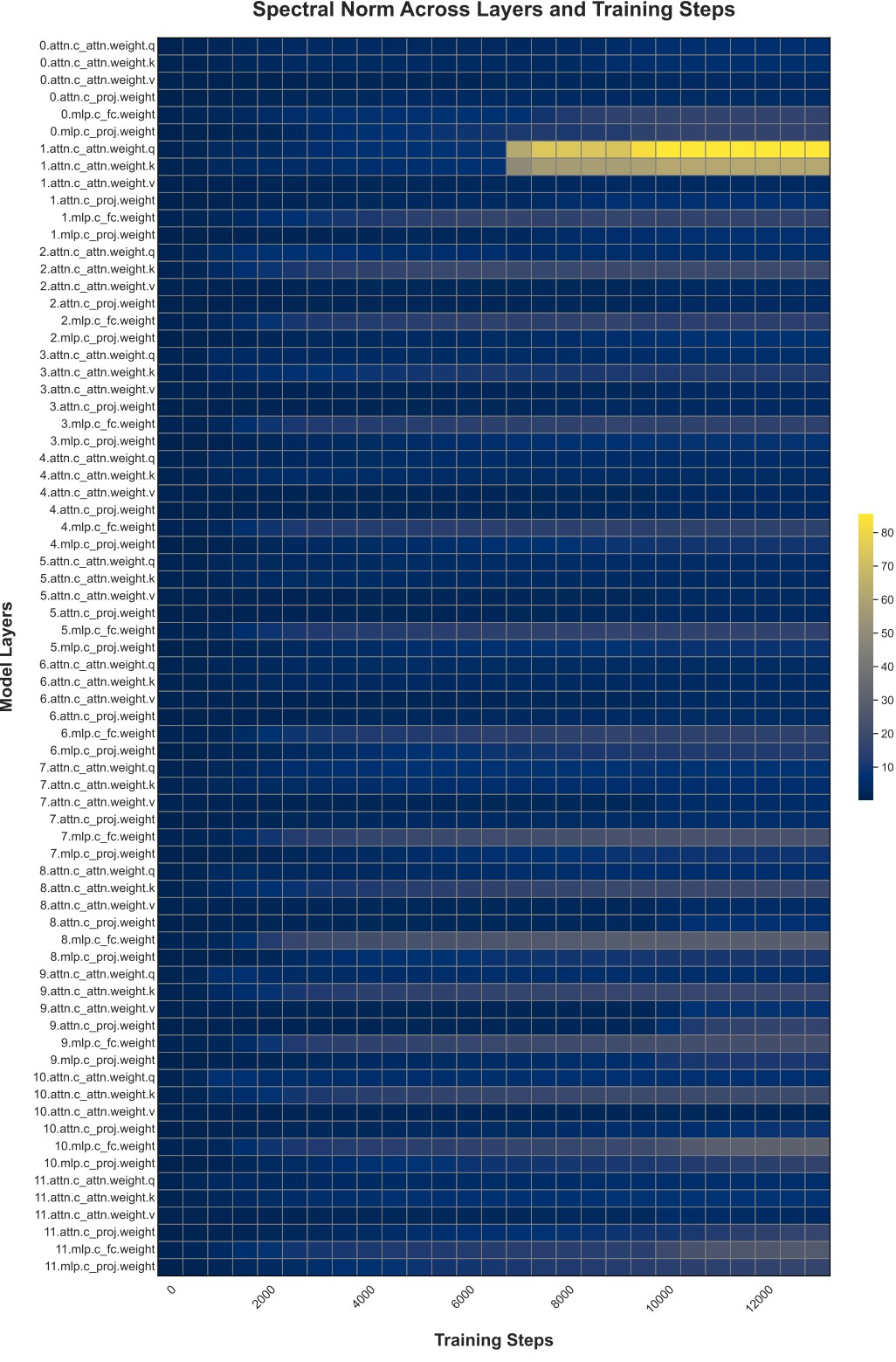

Figure 11: Spectral norm across layers and training steps.

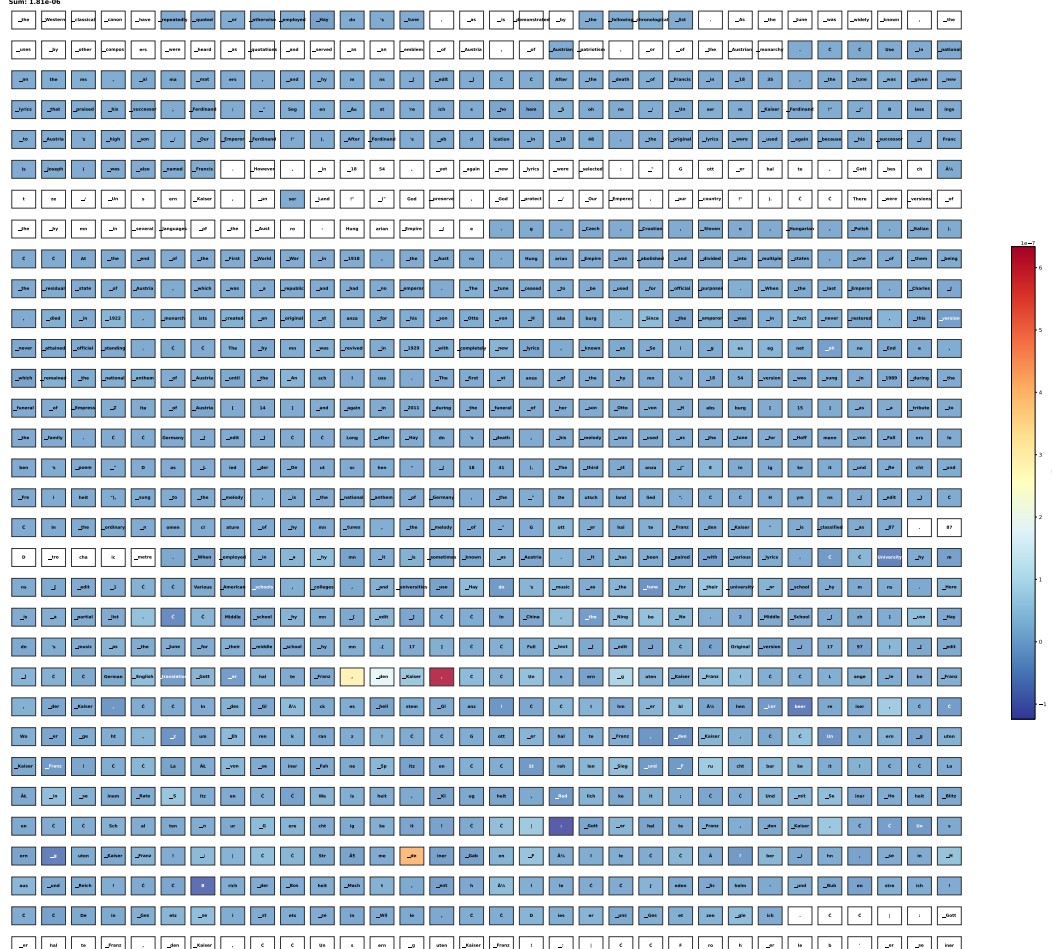

Figure 12: Token difference visualization

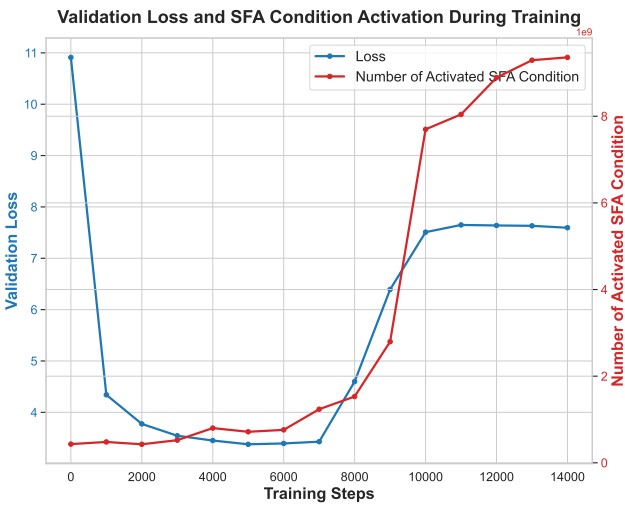

Figure 13: Correlation between multiple maxima occurrences and loss curve.

