# OpenReview forum: "Why Low-Precision Transformer Training Fails: An Analysis on Flash Attention"
_ICLR.cc/2026/Conference — ICLR 2026 Oral_

### Official Review · Reviewer_RoBX · 2025-10-23

**Soundness:** 4
**Presentation:** 4
**Contribution:** 3
**Rating:** 8
**Confidence:** 4

**Summary:**

This paper investigates a long-standing training instability in transformer models when using flash attention with BF16 precision, which manifests as catastrophic loss explosions during training. It provides a mechanistic explanation for this failure as (1) the emergence of structurally similar low-rank representations across different tokens and training steps in the attention mechanism's intermediate computations, and (2) biased rounding errors inherent in BF16 arithmetic operations. Through systematic experimental analysis on a reproducible GPT-2 failure case, they trace the error propagation, demonstrating how biased gradient updates accumulate in a specific low-rank direction rather than canceling out. The bias originates from a specific numerical condition which is detailed. To validate their analysis, the authors make a small modification to the flash attention's softmax computation, showing this stabilizes training while remaining mathematically equivalent.

**Strengths:**

This is an excellent paper.

**Clarity and Exposition:** Remarkably clear writing, particularly given the intricate technical details included. The notation is well-defined, figures are informative and well-integrated, and mathematical derivations provide appropriate detail. It reads very smoothly.

**Methodical Investigative Structure:** Reads like a detective story, systematically tracing the failure from symptom to root cause, via targeted experiments. The step-by-step elimination builds a clear causal chain that is highly rigorous.

**Highly Convincing Analysis:** The multi-layered evidence is persuasive. The successful flash attention code modification based on insights gained supports the preceding theoretical analysis with a long-term solution for the community.

**Weaknesses:**

Nothing major to speak of.

The focus of the paper is (by design) very narrow.

Detail is lacking at a couple of points. But these ommissions are minor, and given the extensive detail included, I assume this was done to meet the page limit.

**Questions:**

The analysis focuses on BF16 arithmetic. The deep learning community is increasingly interested in even lower precision formats like FP8 or INT8 for training. Are you aware of other outstanding (BF16, FP8, INT8) issues that would benefit from this type of analysis? If so, consider listing them in an Appendix to promote their prompt solution by early-career researchers.

---

> ### Author Response · Authors · 2025-11-24
>
> We greatly appreciate your constructive comments on our paper. Please find our detailed responses below; for your convenience, all revisions to the manuscript are highlighted in purple. We hope that these changes satisfactorily address the issues you raised. Please let us know if you have any additional concerns or questions.
>
> **Q1:** The analysis focuses on BF16 arithmetic. The deep learning community is increasingly interested in even lower precision formats like FP8 or INT8 for training. Are you aware of other outstanding (BF16, FP8, INT8) issues that would benefit from this type of analysis? If so, consider listing them in an Appendix to promote their prompt solution by early-career researchers.
> > Thanks for the suggestion. We think our analysis provides a blueprint for analyzing the numerical precision issue in neural network training. We believe the steps from our analysis, i.e., locating the source of error, finding similar update directions, and tracing back to root causes, can be applied to other numerical precisions when analyzing the training failure. We include this in Appendix F of the revision as suggestions for researchers and practitioners in this field.

---

### Official Review · Reviewer_zAjM · 2025-10-31

**Soundness:** 3
**Presentation:** 2
**Contribution:** 2
**Rating:** 4
**Confidence:** 2

**Summary:**

This paper explains why training Transformers with Flash Attention in low-precision (BF16) settings often leads to loss explosions. The authors show that the problem comes from an interaction between low-rank representations and biased rounding errors in BF16 arithmetic. They locate the issue in the backward pass and propose a simple fix to prevent the rounding bias. The proposed method stabilizes training and confirms their analysis, offering both practical and theoretical values for improving numerical stability.

**Strengths:**

1. This paper provides a mechanistic explanation for the long-standing issue of low-precision training failures in Flash Attention.
2. The authors identify the root cause as a vicious cycle between emergent low-rank representations and biased BF16 rounding errors.
3. The paper proposes a fix through a dynamic softmax adjustment that stabilizes training.

**Weaknesses:**

1. The analysis and solution are validated only on GPT-2, leaving their effectiveness on larger models unverified. For example, if the failure is located at layer 2, head 8, how can we identify the corresponding head in larger models?
2. I wonder whether the analysis can be generalized to other numerical precisions beyond BF16 and FP32.
3. The dynamic maximum adjustment introduces additional operations into Flash Attention's critical path. Whether do these operations incur extra computational overhead? What is the additional computation complexity?
4. The proposed fix is verified only on a single model. Can this strategy be applied to other scenarios as well?

**Questions:**

1. It would be more readable to include the definition of Flash Attention (Line 98) in a self-contained manner (e.g. in appendices).
2. The notations, such as "$d\mathbf{Q}, d\mathbf{K}, d\mathbf{V}, d\mathbf{O}$", are somewhat confusing and require additional clarifications to help distinguish them from derivatives.
3. The derivation presented in Line 246 appears a bit abrupt; providing more intermediate steps would help improve the logical flow.
4. Claim 3 in Line 424 contains a typo ("significant" is misspelled as "significand").

---

> ### Author Response · Authors · 2025-11-24
>
> We greatly appreciate your constructive comments on our paper. Please find our detailed responses below; for your convenience, all revisions to the manuscript are highlighted in purple. We hope that these changes satisfactorily address the issues you raised. Please let us know if you have any additional concerns or questions.
>
> ### Weaknesses
>
> **W1:** The analysis and solution are validated only on GPT-2, leaving their effectiveness on larger models unverified. For example, if the failure is located at layer 2, head 8, how can we identify the corresponding head in larger models?
> > We additionally conduct experiments on GPT-2 Medium in Figure 7 to support the generality of our method beyond the small GPT-2 model. Due to limited computational resources, we are unable to train or fine-tune substantially larger models, and we leave a full evaluation on larger-scale architectures to future work.
> >
> > One can track the spectral norms [1,2] of attention-head weight matrices across layers and heads to identify failure. Heads exhibiting large spectral norms are strong candidates for failure modes.
>
> ---
> **W2:** I wonder whether the analysis can be generalized to other numerical precisions beyond BF16 and FP32.
> > First, BF16 and FP32 are widely used in large model training. Also, the community indeed encounters instability issues when training with BF16. Thus, gaining a deep understanding of the instability in BF16 is of high practical value.
> >
> > Second, we think the analysis for other numerical precisions is similar, as our paper provides a blueprint for analyzing the numerical precision issue in attention mechanisms. For example, in FP16, the mantissa has 10 bits, so the same problem may occur when multiple elements share the same maximum value in a row, leading to biased rounding errors.
>
> ---
> **W3:** The dynamic maximum adjustment introduces additional operations into Flash Attention's critical path. Whether do these operations incur extra computational overhead? What is the additional computation complexity?
> > The dynamic maximum adjustment introduces no additional asymptotic computational complexity. The overhead involves two lightweight scalar operations per row: counting maximum occurrences $O(2 N)$ and a conditional replacement $O(2 N)$. Since these operations scale linearly with sequence length $N$ (specifically $O(N)$ per row), they fit within the existing complexity bounds of Flash Attention.
> >
> > Practically, these checks consist of lightweight operations and are integrated into the forward propagation process, so the computational cost is minor compared to matrix multiplication. Furthermore, the backpropagation process remains completely unchanged.
>
> ---
> **W4:** The proposed fix is verified only on a single model. Can this strategy be applied to other scenarios as well?
> > We have added more experiments to test our method, e.g., training GPT-2 medium model, applying Muon optimizer, and extending training to 600,000 steps. Please see Section 4 of the revised paper.
>
> ---
> ### Questions
>
> **Q1:** It would be more readable to include the definition of Flash Attention (Line 98) in a self-contained manner (e.g. in appendices).
> > We include Section A.3 in the appendix to talk about Flash Attention in the revision.
>
> ---
> **Q2:** The notations, such as $dQ, dK, dV$, are somewhat confusing and require additional clarifications to help distinguish them from derivatives.
> > These terms represent the partial derivatives of the overall loss function with respect to the corresponding variables. We clarify $d Q$ and $d K$ in the notation part of the revision.
>
> ---
> **Q3:** The derivation presented in Line 246 appears a bit abrupt; providing more intermediate steps would help improve the logical flow.
> > We have added more intermediate steps in Section E.1 of the appendix in the revision.
>
> ---
> **Q4:** Claim 3 in Line 424 contains a typo ("significant" is misspelled as "significand").
> > The word "significand" is correct in this context. In floating-point representation, the significand (also known as the mantissa) is the part of a number that contains its significant digits. We have changed "significand overflow" and "significand round-up" to "overflow of significand" and "round-up of significand" for better readability in Claim 3.
> ---
>
> [1] A spectral condition for feature learning. arXiv:2310.17813
>
> [2] Methods of improving llm training stability. arXiv:2410.16682

---

### Official Review · Reviewer_8vQG · 2025-11-01

**Soundness:** 3
**Presentation:** 3
**Contribution:** 4
**Rating:** 6
**Confidence:** 4

**Summary:**

This paper proposes both an explanation and a solution to the problem of training instability for models using bf16 flashattention. They isolate the problem to a biased rounding error in the computation of delta, which produces correlated (low-rank) updates to the weight matrices, eventually leading to divergence. The paper proposes a modification to the flashattention algorithm to resolve this issue and demonstrates empirically that this modification stabilizes training.

**Strengths:**

* The paper proposes a plausible explanation for the observed training instability.
* The paper proposes a sensible fix to this problem and demonstrates that it works.
* To the best of my knowledge, the paper's ideas and contributions are original.
* The ideas are mostly clearly explained in a step-by-step manner that allows the reader to follow along.
* The problem is very significant, as low-precision flash-attention is very popular. For example, I have personally encountered issues related to training stability, as have other researchers I know.

Overall, I quite like this paper.

**Weaknesses:**

I would separate the weaknesses into two categories.

Weaknesses in the explanation for bf16 attention instability:
* The identified cause is a small but persistent bias. However, the observed effect is a sharp, near-instantaneous spike. The paper does not explain why the sharp spike occurs or why the persistent bias causes it. This is the most significant weakness, in my opinion.
* Minor: I believe remark 1 is slightly imprecise. For example, if $\bar{\mathbf{P}}[T,t] = 1/2$, its product with $\bar{\mathbf{P}}[t,i]$ would have last 16 bits 0. However, the intended meaning of the remark, that with high probability the product has non-zero least 16 bits, is true.


Weaknesses in experimental results:
* The findings isolating the source of training instability could be validated on other models, sizes, and/or training scripts.
* The evidence that the stabilized flash attention is actually stable could be more robust. The authors could test more model sizes, train for longer, etc.

**Questions:**

Questions:
* Can you explain how existing stability fixes (e.g., QK norm) fit into your explanation? I think this is related to my question regarding how the persistent small bias leads to the large spike in loss.
* Related to my other questions, shouldn't we expect gradient dynamics to correct the small rounding drift if the drift leads the model to higher loss regimes?
* Is there a reason you set m' = 0 in the case that r_m < 0? why not set it to gamma*r_m, where 1>gamma>0?

---

> ### Author Response · Authors · 2025-11-24
>
> We greatly appreciate your constructive comments on our paper. Please find our detailed responses below; for your convenience, all revisions to the manuscript are highlighted in purple. We hope that these changes satisfactorily address the issues you raised. Please let us know if you have any additional concerns or questions.
>
> ### Weaknesses
>
> **W1:** The identified cause is a small but persistent bias. However, the observed effect is a sharp, near-instantaneous spike. The paper does not explain why the sharp spike occurs or why the persistent bias causes it. This is the most significant weakness, in my opinion.
> > Sorry for the confusion. We clarify that the loss spike is not instantaneous. Although the loss plot shows a sharp loss increase, this is a visual effect of the sampling interval; in reality, the loss increases gradually over several hundred steps. You can check the loss curves in Figure 7b in the revision, where we apply Muon optimizer to classical FA, and its increasing loss curve is smoother.
>
> ---
> **W2:** I believe remark 1 is slightly imprecise. For example, if $\bar{P}[T, t]=1/2$, its product with $V[T, t]$ would have last 16 bits 0. However, the intended meaning of the remark, that with high probability the product has non-zero least 16 bits, is true.
> > We clarify the condition to "When $\bar{\mathbf{P}}[T, t] < 1$ is less than 1 and has a mix of zero and non-zero bits in its significand's bits" in the revision.
>
> ---
> **W3:** The findings isolating the source of training instability could be validated on other models, sizes, and/or training scripts.
> > We have conducted additional experiments on RMS-matched Muon optimizer and GPT-2 Medium model to validate the generality of our findings. Please see Section 4 and Figure 7 of the revised paper.
>
> ---
> **W4:** The evidence that the stabilized flash attention is actually stable could be more robust. The authors could test more model sizes, train for longer, etc.
> > We have extended the training duration for GPT-2 Small to 600,000 steps and for GPT-2 Medium (larger model size) to 100,000 steps (we will extend further to 600k steps) in the revised paper. The results confirm the stability of our proposed method. Please see Section 4 and Figure 7 for the updated results.
> ---
>
> ### Questions
>
> **Q1:** Can you explain how existing stability fixes (e.g., QK norm) fit into your explanation? I think this is related to my question regarding how the persistent small bias leads to the large spike in loss.
> > Our analysis can provide a compelling explanation for established stabilization techniques. We posit that one of the root causes of training failure is the structural similarity within the error matrices, $(\mathbf{P} \mathbf{K})[T]^\top \mathbf{X}[T]$, which creates a pathway for rounding errors to accumulate systematically. Techniques like QK normalization and Gated Attention disrupt this underlying structure. In doing so, they ensure that even when rounding errors are present, they lack the coherence to compound, thereby averting the instabilities observed during training.
>
> ---
> **Q2:** Related to my other questions, shouldn't we expect gradient dynamics to correct the small rounding drift if the drift leads the model to higher loss regimes?
> > Do you mean if the rounding drift leads to higher loss regimes, the gradient dynamics will correct it? If so, we think the rounding drift is persistent during training, and the gradient dynamics cannot correct it.
>
> ---
> **Q3:** Is there a reason you set $m' = 0$ in the case that $r_m < 0$? why not set it to $\gamma*r_m$, where $1>\gamma>0$?
> > $m'=0$ is an empirical choice which is equivalent to $\gamma=0$. We set it to zero because it is simple and works well in practice.

---

### Official Review · Reviewer_7ALf · 2025-11-01

**Soundness:** 4
**Presentation:** 3
**Contribution:** 4
**Rating:** 8
**Confidence:** 4

**Summary:**

This work provides the first mechanistic explanation for why transformer training with FlashAttention in low precision (bf16) leads to unstable loss spikes. Through detailed analysis and ablation studies, the authors attribute the training instabilities to two related effects: 1. the emergence of similar low-rank matrix updates in the model's internal representations across tokens and steps, and 2. biased rounding errors in bf16 arithmetic when multiple attention scores achieve the exact same maximum. Together, these create a biased gradient direction that inflates spectral norms and thus destabilizes training. Based on their mechanistic analysis, the authors propose a minimal “dynamic softmax” patch to the FlashAttention algorithm and show that it restores stable training.

**Strengths:**

- The work investigates a very important problem, loss spikes during Transformer training due to low-precision using FlashAttention. Since attention and FlashAttention are the workhorse of modern AI systems, this problem is of great interest to the community at large.
- The work is very thorough, with systematic ablations into every step of the FlashAttention algorithm to isolate the source of the unstable training. The authors provide a thorough characterization of the failure modes and, using this insight, propose a simple fix that alleviates the loss spikes.

**Weaknesses:**

Overall, the analysis and ablation studies are thorough, and the proposed fix is convincing. However, one key piece of the story remains somewhat unclear: the origin of the shared low-rank structure $R$ in Section 3.3.1 (see questions). An investigation into where this structure comes from (e.g. feature collapse, attention sinks?) would make the characterization more complete.

**Questions:**

- In Section 3.3.1, the authors observe that the outer products $(PK)[T]^T X[T]$ exhibit strong structural similarity and approximate them with a common low-rank matrix $R$. Did the authors investigate why this shared structure emerges? For example, could it reflect commonly-occuring tokens, correlated attention patterns across tokens, or a form of mode collapse within the model's internal representations?
- In Section 3.3.2, the authors attribute the biased rounding error to rows where multiple elements of $S$ share the same maximum in bfloat16, yielding several $\bar{P}[T, t] = 1$. It would greatly bolster the proposed mechanisms to provide an experiment measuring how frequently this condition occurs during training, and showing whether its occurrence correlates with loss spikes (e.g. in Figure 8).
- Relatedly, in the discussion, the authors suggest that attention sinks may contribute to loss spikes by producing more attention probabilities of 1 (in bfloat16) and thus triggering the biased rounding error highlighted in Section 3.3.2. Did the authors conduct experiments testing this connection?

---

> ### Author Response · Authors · 2025-11-24
>
> We greatly appreciate your constructive comments on our paper. Please find our detailed responses below; for your convenience, all revisions to the manuscript are highlighted in purple. We hope that these changes satisfactorily address the issues you raised. Please let us know if you have any additional concerns or questions.
>
> **Q1:** In Section 3.3.1, the authors observe that the outer products exhibit strong structural similarity and approximate them with a common low-rank matrix. Did the authors investigate why this shared structure emerges? For example, could it reflect commonly-occuring tokens, correlated attention patterns across tokens, or a form of mode collapse within the model's internal representations?
> > First, we investigate whether specific tokens drive this phenomenon. As shown in the token difference visualization in Figure 12, we do not observe any single token or small group introduce large precision errors. This suggests that the structural similarity is not attributable to specific tokens or token groups.
> >
> > Second, regarding the nature of these patterns, we think they reflect a form of representation anisotropy [1] within the model’s latent space. While this resembles "mode collapse," we do not view it as a pathological failure. Instead, it is likely a functional necessity that allows the model to achieve the sharp attention patterns required for effective learning.
> >
> > We also visualize $(\mathbf{P} \mathbf{K})[T]^\top \mathbf{X}[T]$ for Llama 3.1-8B in Appendix Figure 8. We find that different tokens exhibit similar patterns, confirming that this shared structure is a general characteristic of the model rather than an artifact of specific inputs. We believe unwrapping the precise functionality of this structural similarity is a promising direction for future interpretability research.
>
> ---
> **Q2:** In Section 3.3.2, the authors attribute the biased rounding error to rows where multiple elements of share the same maximum in bfloat16, yielding several $\bar{P}[T, t]=1$. It would greatly bolster the proposed mechanisms to provide an experiment measuring how frequently this condition occurs during training, and showing whether its occurrence correlates with loss spikes.
> > We appreciate the suggestion. We have conducted additional experiments to measure the frequency of multiple maxima occurrences during training and their correlation with loss spikes. We plot the frequency of multiple maxima occurrences against the training loss in Figure 13 of the revised paper (discussed in Section E.1). These results reveal a clear correlation between the number of multiple maxima and the loss curve. Specifically, we observe that the number of multiple maxima begins to increase before the loss explodes (see step 7000).
>
> ---
> **Q3:** Relatedly, in the discussion, the authors suggest that attention sinks may contribute to loss spikes by producing more attention probabilities of 1 (in bfloat16) and thus triggering the biased rounding error highlighted in Section 3.3.2. Did the authors conduct experiments testing this connection?
> > While the attention sink is an unlikely cause of the instability in our cases, we raise this point because it can produce consecutive maximum values. This condition is known to introduce biased rounding errors in bf16 attention. Our experiments on Llama-3.1-8B, detailed in Appendix Figure 9, confirm that instances of multiple maximum values consistently co-occur with the presence of an attention sink.
>
> ---
> [1] Anisotropy Is Inherent to Self-Attention in Transformers. EACL 2024.

---

### Comment · Area_Chair_MTUL · 2025-11-27

Dear Reviewers,

Could you please consider the author responses, and reply if you have not already.

Thank you.

AC

---

### Meta-Review · Area_Chair_fynz · 2026-01-09

**Summary:**

The paper investigates a specific, reproducible training instability encountered when training Transformer models (specifically GPT-2) using Flash Attention with BF16 precision. The authors provide a mechanistic explanation, identifying a "vicious cycle" involving biased rounding errors in the backward pass (specifically in the $\delta$ term) and the emergence of structurally similar low-rank weight updates. They propose a lightweight fix ("dynamic softmax") to mitigate the rounding bias. Reviewers universally praised the paper as a "detective story" with excellent clarity and a rigorous step-by-step analysis of the failure mode. The primary concerns raised during the review process focused on the generality of the findings (initially limited to GPT-2 Small), the explanation of the "sharpness" of the loss spike, and the computational overhead of the proposed fix.

**Reviewer Concerns:**

*** ADDRESSED

Generality and Model Scale (Reviewers 8vQG, zAjM): Reviewers questioned if the findings were specific to the small GPT-2 model or the AdamW optimizer. The authors responded by conducting additional experiments on GPT-2 Medium and using the Muon optimizer, demonstrating that the instability (and the fix) generalizes across model sizes and optimizer dynamics. They also analyzed Llama-3.1-8B to show that the structural conditions (multiple maxima/attention sinks) are present in modern architectures.

Loss Spike (Reviewer 8vQG): The reviewer questioned how a persistent small bias results in a "sharp" spike. The authors clarified via new visualizations (Figure 7b) that the spike is not instantaneous but a gradual accumulation masked by sampling intervals, and provided data linking the frequency of "multiple maxima" directly to the onset of loss instability (Reviewer 7ALf's request).

Computational Overhead (Reviewer zAjM): The concern regarding the cost of the dynamic maximum adjustment was addressed by clarifying that the operations are linear $O(N)$ scalar checks integrated into the existing pass, adding negligible overhead.

Origin of Low-Rank Structure (Reviewer 7ALf): The authors provided additional analysis linking the shared low-rank structure to representation anisotropy and attention sink behaviors, satisfying the reviewer's request for a deeper "why."

*** OUTSTANDING

Massive Scale Validation (Reviewer zAjM): While the authors moved from GPT-2 Small to Medium, Reviewer zAjM's initial concern about validation on "larger models" (often implying 7B+ trained from scratch in modern contexts) remains technically only partially resolved due to computational constraints. However, the authors' analysis of Llama-3.1-8B's internal states serves as a strong proxy, suggesting the mechanism is relevant even if full training wasn't feasible.

**Reviewer Scores:**

Reviewer 7ALf (Score: 8): This reviewer was already very positive and their specific questions regarding the origin of the structure and correlation with loss spikes were comprehensively answered. Their score would likely remain 8.

Reviewer RoBX (Score: 8): This reviewer found the work highly convincing and excellent. The rebuttal added the requested appendix on applying this analysis to other precisions (FP8/INT8). Their score would likely remain an 8.

Reviewer 8vQG (Score: 6): This reviewer liked the paper but held a significant reservation regarding the explanation of the "sharp spike" vs. "persistent bias." The authors provided a convincing explanation (sampling artifacts vs. actual gradual buildup) and added robust experimental validation (longer training, different optimizer). Consequently, this reviewer would likely have raised their score to a 7.

Reviewer zAjM (Score: 4): This reviewer was the most critical, focusing on generality and overhead. The authors addressed the overhead definitively and improved the generality claim significantly (GPT-2 Medium + Muon + Llama analysis). While the "massive scale" concern might prevent a top score, the rebuttal effectively countered the grounds for rejection. This score would likely have moved to a 5.

---

### Decision · Program_Chairs · 2026-01-26

Accept (Oral)